# Lymphangiogenic therapy prevents cardiac dysfunction by ameliorating inflammation and hypertension

LouJin Song[1], Xian Chen[2,3], Terri A Swanson[4], Brianna LaViolette[2], Jincheng Pang[1], Teresa Cunio[1,5†], Michael W Nagle[1,6‡], Shoh Asano[1], Katherine Hales[1], Arun Shipstone[7], Hanna Sobon[7], Sabra D Al-Harthy[2,3], Youngwook Ahn[8], Steven Kreuser[2], Andrew Robertson[3], Casey Ritenour[3], Frank Voigt[3], Magalie Boucher[3], Furong Sun[4], William C Sessa[9], Rachel J Roth Flach[1]*

[1]Internal Medicine Research Unit, Pfizer Inc, Cambridge, United States; [2]Comparative Medicine, Pfizer Inc, Cambridge, United States; [3]Drug Safety Research & Development, Pfizer Inc, Groton, United States; [4]Early Clinical Development, Pfizer Inc, Cambridge, United States; [5]Acceleron Pharma, Cambridge, United States; [6]Eisai Inc, Cambridge, United States; [7]Inflammation and Immunology Research Unit, Pfizer Inc, Cambridge, United States; [8]Target Sciences, Emerging Science and Innovation, Pfizer Inc, Cambridge, United States; [9]Department of Pharmacology, Vascular Biology and Therapeutics Program, Yale University School of Medicine, New Haven, United States

*For correspondence:
Rachel.RothFlach@pfizer.com

Present address: †Acceleron Pharmaceuticals; ‡Eisai Pharmaceuticals

**Abstract** The lymphatic vasculature is involved in the pathogenesis of acute cardiac injuries, but little is known about its role in chronic cardiac dysfunction. Here, we demonstrate that angiotensin II infusion induced cardiac inflammation and fibrosis at 1 week and caused cardiac dysfunction and impaired lymphatic transport at 6 weeks in mice, while co-administration of VEGFCc156s improved these parameters. To identify novel mechanisms underlying this protection, RNA sequencing analysis in distinct cell populations revealed that VEGFCc156s specifically modulated angiotensin II-induced inflammatory responses in cardiac and peripheral lymphatic endothelial cells. Furthermore, telemetry studies showed that while angiotensin II increased blood pressure acutely in all animals, VEGFCc156s-treated animals displayed a delayed systemic reduction in blood pressure independent of alterations in angiotensin II-mediated aortic stiffness. Overall, these results demonstrate that VEGFCc156s had a multifaceted therapeutic effect to prevent angiotensin II-induced cardiac dysfunction by improving cardiac lymphatic function, alleviating fibrosis and inflammation, and ameliorating hypertension.

## Introduction

The lymphatic vasculature plays an important role in mediating fluid balance and local inflammation in many tissues and organs throughout the body (*Huang et al., 2017*; *Oliver et al., 2020*). The heart also carries a complex lymphatic network and relies on cardiac lymphatics to drain interstitial fluids and immune cells for maintaining homeostasis (*Brakenhielm and Alitalo, 2019*). Disrupting cardiac lymphatics can cause myocardial edema and lead to impaired cardiac function (*Cui et al., 2001*; *Mehlhorn et al., 1995*), indicating a critical role of cardiac lymphatics in cardiovascular physiology. Recently, cardiac lymphatics have emerged as a new therapeutic target for acute cardiac injuries. Several studies have demonstrated using rodent models that therapeutic lymphangiogenesis

reduced myocardial edema and fibrosis, resolved inflammation, and improved cardiac function after myocardial infarction (*Henri et al., 2016*; *Houssari et al., 2020*; *Klotz et al., 2015*; *Tatin et al., 2017*; *Trincot et al., 2019*; *Vieira et al., 2018*) or ischemia and reperfusion injury (*Shimizu et al., 2018*). These studies in aggregate concluded that the improvement in cardiac function was due to local proliferation of lymphatic vessels at the site of injury in response to lymphangiogenic therapies. Importantly, these animal models represent cardiac injuries, in which blood and lymphatic vessels are surgically manipulated. However, the role of the lymphatic vasculature in the pathogenesis of chronic cardiac dysfunction and the effects of lymphangiogenic therapies on cardiac dysfunction remain to be explored. Here, we explored the role of the lymphatic vasculature in cardiac dysfunction and investigated the effects of a lymphangiogenic therapy (VEGFCc156s) on the angiotensin II infusion-induced cardiac dysfunction model. We found that angiotensin II infusion-induced cardiac dysfunction is associated with impaired lymphatic vascular function as well as increased cardiac fibrosis and inflammation, whereas VEGFCc156s treatment ameliorated these pathologies. Interestingly, angiotensin II-infused mice treated with VEGFCc156s also demonstrated a delayed and dramatic response to reverse angiotensin II-induced hypertension. Thus, we propose that the beneficial effects of the lymphangiogenic therapy VEGFCc156s on angiotensin II infusion-induced cardiac dysfunction are multifaceted and mediated by its concomitant effects on cardiac lymphatic function, inflammatory and fibrotic gene expression, and systemic hypertension.

## Results

### Lymphatic endothelial markers associate with cardiac comorbidities and are reduced in human failing hearts

To investigate the role of the lymphatic vasculature in human heart failure and cardiac dysfunction, we first analyzed human genome wide association studies (GWAS) for single-nucleotide polymorphisms (SNPs) that were near lymphatic marker gene loci and were associated with human cardiovascular or metabolic comorbidities. Interestingly, several SNPs near lymphatic endothelial markers *VEGFC*, *LYVE1*, *FLT4* loci are associated with cardiovascular and metabolic phenotypes including cardiac hypertrophy in humans (*Fung et al., 2019*; *Johansson et al., 2010*; *justice et al., 2019*; *Kettunen et al., 2016*; *Kichaev et al., 2019*; *Lotta et al., 2018*; *Lu et al., 2017*; *Nasa Sinnott-Armstrong et al., 2019*; *Saxena et al., 2010*; *van der Harst and Verweij, 2018*; *Wild et al., 2017*; *Xu et al., 2018*; *Table 1*). In addition, human heart samples were obtained from healthy donors or patients with chronic heart failure, and the gene expression of lymphatic endothelial markers *VEGFC*, *LYVE1*, *PDPN*, and *FLT4* was assessed. A significant (19.6% for *VEGFC*, 38.8% for *LYVE1* and 21.7% for *FLT4* on average) reduction was observed for lymphatic markers *VEGFC*, *LYVE1*, and *FLT4*, and a 31.6% reduction was observed in *PDPN* ($p$=0.07) in hearts from chronic heart failure patients compared with hearts from healthy donors (*Figure 1A–D*, *Figure 1—source data 1*), suggesting that the cardiac lymphatic vasculature is impaired in human chronic heart failure.

### VEGFCc156s prevented angiotensin II-induced cardiac dysfunction

To examine the role of the lymphatic vasculature in cardiac dysfunction further, we used a model of angiotensin II infusion, which promotes cardiac dysfunction directly via its effects on cardiomyocytes (*Mazzolai et al., 2000*; *Mehta and Griendling, 2007*) and indirectly by its action on the renin-angiotensin system to induce hypertension (*Mehta and Griendling, 2007*). To induce lymphangiogenesis in this model, we used the lymphangiogenic growth factor mutant VEGFCc156s, which specifically activates vascular endothelial growth factor receptor 3 (VEGFR3) but not vascular endothelial growth factor receptor 2 (VEGFR2) in vitro (*Joukov et al., 1998*) and enhances lymphangiogenesis in vivo (*Klotz et al., 2015*). Pharmacokinetic analysis of single subcutaneous dosing of VEGFCc156s indicated that VEGFCc156s has a short half-life in plasma (*Figure 2—figure supplement 1A*). Thus, we hypothesized that continuous infusion to maintain a stable increase in plasma VEGFCc156s concentration would be the most effective mode of administration. Therefore, an osmotic pump was implanted subcutaneously to infuse angiotensin II and VEGFCc156s simultaneously (*Figure 2A*). The VEGFCc156s infusion was calculated to maintain a two-fold increase in plasma VEGFC concentration, and the analysis at endpoint confirmed a significant 1.9-fold average increase in plasma VEGFC concentration in the treated mice compared with control mice (*Figure 2—figure supplement 1B*).

**Table 1.** SNPs mapping to lymphatic endothelial markers associate with cardiac comorbidities.

SNPs in human genetic loci, where lymphatic endothelial marker genes *VEGFC*, *LYVE1*, *FLT4* are located, are associated with cardio-vascular and metabolic phenotypes. *MRVI1*, murine retrovirus integration site one homolog. *ADM*, adrenomedullin. *CTR9*, RNA poly-merase-associated protein CTR9 homolog. *MGAT1*, alpha-1,3-mannosyl-glycoprotein 2-beta-N-acetylglucosaminyltransferase. *SCGB3A1*, Secretoglobin Family 3A Member 1. *OR2Y1*, olfactory receptor family two subfamily Y member 1.

| SNP | Nearest gene(s) | Associated phenotype | p Value | Reference |
|---|---|---|---|---|
| rs2333496 | VEGFC | Waist-to-hip ratio (WHR) adj BMI | 8.00E-11 | Lotta, et al. |
| rs7660760 | VEGFC | Left ventricle wall thickness | 4.00E-07 | Wild, et al. |
| rs309795 | VEGFC | 2 hr Glucose Adj. for BMI | 2.22E-06 | Saxena, et al. |
| rs114108584 | VEGFC | Idiopathic dilated cardiomyopathy | 9.00E-06 | Xu, et al. |
| rs11042906 | LYVE1 and MRVI1 | Systolic BP | 2.00E-14 | Kichaev, et al. |
| rs2218793 | LYVE1 and ADM | High density lipoprotein cholesterol (HDL-C) | 1.60E-11 | Sinnott-Armstrong, et al. |
| rs11042937 | LYVE1 and CTR9 | Coronary artery disease | 3.00E-10 | Van der harst, et al. |
| rs7940646 | LYVE1 and MRVI1 | Triglycerides 2017 Mostly European and East Asian | 2.93E-08 | Lu, et al. |
| rs10840457 | LYVE1 and MRVI1 | Arterial stiffness index | 3.00E-08 | Fung, et al. |
| rs11603178 | LYVE1 and MRVI1 | Diastolic BP | 9.85E-08 | Nealelab-uk-biobank |
| rs12807023 | LYVE1 and ADM | Plasma + serum IDL-TAG levels | 8.21E-07 | Kettunen, et al. |
| rs12807023 | LYVE1 and ADM | Plasma + serum XS-VLDL values (concentration, TAG) | 5.41E-06 | Kettunen, et al. |
| rs634501 | FLT4 and MGAT1 | High density lipoprotein cholesterol (HDL-C) | 9.35E-08 | Lu, et al. |
| rs142342609 | FLT4 and SCGB3A1 | WHR adjusted BMI | 7.00E-07 | Justice, et al. |
| rs12517906 | FLT4 and OR2Y1 | Body mass index (females) | 6.00E-06 | Johansson, et al. |

Echocardiography was conducted at endpoint to examine cardiac function, and the results indicated that angiotensin II infusion induced cardiac hypertrophy and impaired cardiac contractility. In contrast, VEGFCc156s treatment prevented angiotensin II-induced impairments in cardiac function (*Figure 2B–E*, *Figure 2—figure supplement 1C–1G*). Consistent with the echocardiography results, angiotensin II infusion also significantly increased heart weight and lung weight, which possibly reflects an increase in lung edema (*King and Goldstein, 2020*), whereas VEGFCc156s prevented the angiotensin II-induced increases in heart and lung weights (*Figure 2F and G*). Moreover, gene expression analysis revealed a significant increase in the expression of *Nppb*, encoding heart failure marker B-type natriuretic peptide (BNP), in angiotensin II-treated mouse hearts compared with control hearts, while VEGFCc156s significantly decreased *Nppb* expression in angiotensin II-infused hearts (*Figure 2H*).

Finally, cardiomyocyte cell size was examined using wheat germ agglutinin (WGA), which stains the cell membrane. Whereas angiotensin II infusion significantly increased cardiomyocyte size and heterogeneity, VEGFCc156s treatment prevented the change in cell size heterogeneity in angiotensin II-infused mouse hearts (*Figure 2—*

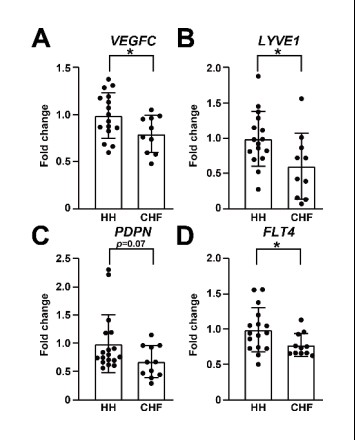

**Figure 1.** Lymphatic endothelial markers are reduced in human failing heart. (**A–D**) Expression of lymphatic endothelial marker genes *VEGFC* (**A**), *LYVE1* (**B**), *PDPN* (**C**), *FLT4* (**D**) in hearts from healthy human donors (HH) and hearts from patients with chronic heart failure (CHF). The data were normalized to the expression of the housekeeping gene *HPRT1* and represented as fold change from the HH group (*n* = 16–18 for HH and *n* = 10–11 for CHF). Data are mean ± s.d. Student's *t*-test was used for statistics. \**p*<0.05.

The online version of this article includes the following source data for figure 1:

**Source data 1.** Clinical information.
**Source data 2.** Raw data supporting *Figure 1*.

figure supplement 1H–1J), confirming the beneficial effects of VEGFCc156s on angiotensin II-induced cardiac hypertrophy.

To understand how VEGFCc156s ameliorated cardiac dysfunction, the effect of VEGFCc156s on cardiac lymphatic vascular marker expression and VEGFR3-mediated signaling cascades was examined. Similar to what was observed in human hearts (Figure 1), a significant reduction in lymphatic endothelial marker podoplanin expression was observed in angiotensin II-infused hearts compared with control hearts, indicating a potential reduction in cardiac lymphatic vascular density. Interestingly, VEGFCc156s treatment restored podoplanin expression in angiotensin II-infused hearts back to control level, indicating that VEGFCc156s treatment may prevent the loss of cardiac lymphatic vessels during cardiac dysfunction (Figure 2—figure supplement 1K and L). Consistent with the protein expression analysis, immunohistochemistry revealed a significant reduction in podoplanin-positive lymphatic vessel density in angiotensin II-infused hearts, which was restored by VEGFCc156s treatment (Figure 2I and J). More interestingly, podoplanin-positive lymphatic vessel lumen area was significantly increased in the hearts of VEGFCc156s + angiotensin II-infused animals compared with animals that had been treated with angiotensin II alone, indicating that VEGFCc156s may improve lymphatic transport function in diseased animals (Figure 2K). Consistent with the ability of VEGFC to activate Akt and Erk pathways, we observed increased phospho-Akt levels and a trend towards an increase in phospho-Erk in VEGFCc156s-treated hearts compared with control or angiotensin II-infused hearts, indicating the activation of signaling pathways that promote survival and growth by VEGFCc156s treatment (Figure 2—figure supplement 1K, M and N). To further characterize the systemic lymphangiogenic effect of VEGFCc156s, we examined lymphatic vessel diameter and density in the ear skin by immunohistochemistry. Similar to its effects on cardiac lymphatic vessels, VEGFCc156s treatment also significantly increased lymphatic vessel diameter and lymphatic vascular density in the ear skin of treated animals, supporting a systemic lymphangiogenic response induced by VEGFCc156s treatment (Figure 2—figure supplement 1O–1R).

## VEGFCc156s improved cardiac lymphatic vascular function

To investigate whether the changes in podoplanin-positive lymphatic vessel density and lumen area in heart were translated into functional changes, we conducted lymphangiography to directly evaluate cardiac lymphatic transport function. FITC-dextran and cadaverine were injected intravenously to label the cardiac blood vasculature and reveal any vascular leak, respectively. Next, fluorescent qdots were injected into the apex of the heart. Qdots are macromolecules that diffuse into interstitial fluid and may be cleared by the lymphatic vasculature and transported back to the blood circulation (Henri et al., 2016; Shimizu et al., 2018). Angiotensin II treatment induced vascular leakage, leading to interstitial accumulation of cadaverine in angiotensin II-infused hearts. In contrast, VEGFCc156s-treated, angiotensin II-infused animals showed significantly reduced cadaverine accumulation in heart, indicating reduced leakage and improved fluid clearance (Figure 3A and B, Figure 3—figure supplement 1). We observed that in all animals, the qdots were transported back to the blood circulation within 5 mins after injection. However, quantification of the plasma qdot concentration, which is an indirect parameter to evaluate cardiac lymphatic transport efficiency, demonstrated a significant decrease in angiotensin II-infused mice compared with control mice, indicating impaired cardiac lymphatic transport. Conversely, VEGFCc156s treatment in the angiotensin II-infused mice significantly increased the plasma qdot concentration compared with the angiotensin II-infused mice, indicating improved cardiac lymphatic transport (Figure 3C). Concomitantly, a trend toward an increased heart wet-to-dry weight ratio was observed in angiotensin II-infused hearts, which was significantly reduced with VEGFCc156s treatment, indicating improved regulation of cardiac fluid balance in VEGFCc156s-treated mice (Figure 3D). These results demonstrated that VEGFCc156s treatment improved cardiac lymphatic vascular function in angiotensin II-infused animals, which may facilitate fluid balance regulation and contribute to the improved cardiac function observed after VEGFCc156s treatment.

## VEGFCc156s ameliorated angiotensin II-induced inflammatory responses

To further characterize the effects of VEGFCc156s on angiotensin II infusion-induced cardiac dysfunction, we conducted RNA sequencing (RNA-seq) analysis to profile the transcriptional changes in

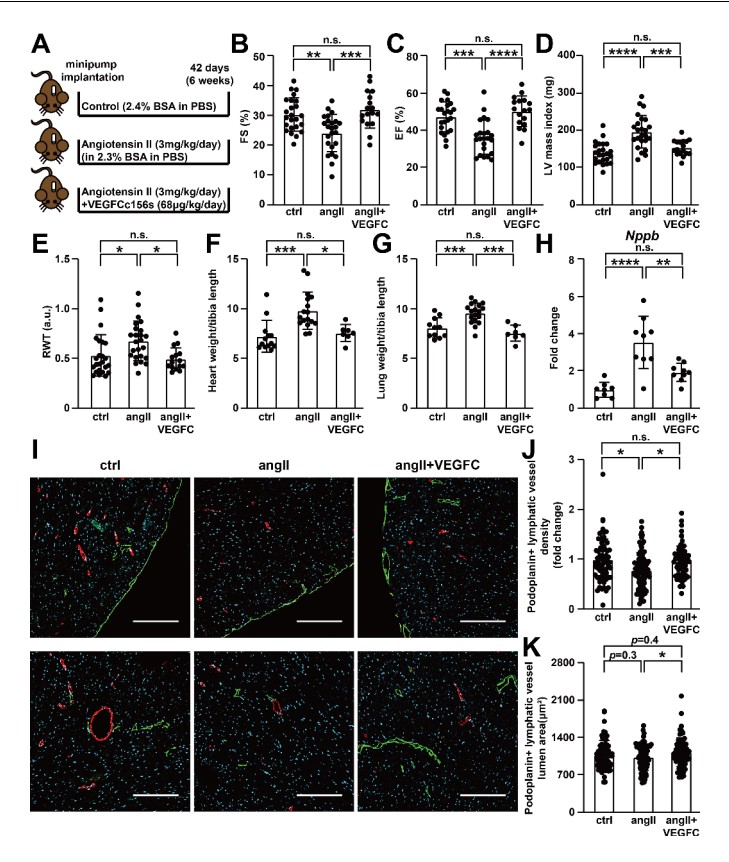

**Figure 2.** VEGFC$_{C156S}$ treatment prevented angiotensin II-induced cardiac dysfunction. (**A**) Experimental design for animal studies. Angiotensin II (angII) was infused to induce cardiac dysfunction, and VEGFCc156s (VEGFC) was infused as a lymphangiogenic therapy via the same subcutaneously-implanted osmotic pump. Bovine serum albumin (BSA) was loaded to the pumps for control (ctrl) and angII groups to balance for the loading of VEGFCc156s in the therapeutic arm. (**B–E**) Echocardiography was conducted 6 weeks after minipump implantation. Fractional Shortening (FS) (**B**), Ejection Fraction (EF) (**C**), Left Ventricular (LV) mass index (**D**) and Relative Wall Thickness (RWT) (**E**) are shown (n = 17–24/group). (**F–G**) Heart weight (**F**) and lung weight (**G**) (normalized to tibia length) of the animals at euthanasia (n = 7–18/group). (**H**) RNA was isolated from the mouse hearts, and quantitative RT-PCR was performed for *Nppb* and normalized to the expression of housekeeping gene *Hprt*. The data were normalized to ctrl group and represented as fold change (n = 8–9/group). (**I**) Mouse hearts were fixed, OCT embedded, sectioned, and stained for podoplanin, smooth muscle actin and DAPI (light blue). Representative images of podoplanin-positive lymphatic vessels (green) and smooth muscle actin-positive arteries (red) for each group were shown. Scale bar, 200 µm. (**J**) Quantification of podoplanin-positive lymphatic vessel density (**J**, normalized to total area and represented as fold change to ctrl group) and podoplanin-positive lymphatic vessel lumen area (**K**) in mouse hearts (n = 8–10 animal/group, n = 10–12 sections/animal). Data are mean ± s.d. One-way *ANOVA* with Bonferroni posthoc was used for all figures except 2J and 2K (a linear mixed model was used for statistics). *p<0.05, **p<0.01, ***p<0.001, ****p<0.0001, n.s. not significant.
The online version of this article includes the following source data and figure supplement(s) for figure 2:

**Source data 1.** Raw data supporting *Figure 2*.
**Figure supplement 1.** Effect of VEGFC$_{C156S}$ on angiotensin II-induced cardiac dysfunction, cardiac lymphatics, and skin lymphatics.

different cell populations from the heart. Transcriptional changes in diaphragm lymphatic endothelial cells (LECs) were also profiled to evaluate the effect of VEGFCc156s on peripheral lymphatics, as the treatment is systemic. To efficiently distinguish LECs from blood endothelial cells (BECs), we generated a Prox1-eGFP knock-in mouse model in which eGFP expression is driven by the endogenous promoter of the lymphatic endothelial marker *Prox1* gene using CRISPR/Cas9 technology and confirmed the eGFP signals from lymphatic vessels in the heart and the diaphragm (*Figure 4—figure*

*supplement 1A–1C*). The Prox1-eGFP mice were next used for in vivo studies following the study design shown in *Figure 2A*. From the animals, LECs, BECs and cardiomyocytes were isolated from the heart, and LECs were additionally isolated from the diaphragm following an optimized protocol (*Figure 4—figure supplement 1D*) and used for RNA-seq. A principal component analysis of the normalized expression counts across cell types indicated that the samples clustered based on cell population and showed a distinct expression profile (*Figure 4—figure supplement 1E*). Differential expression and signaling pathway analysis revealed that angiotensin II significantly altered gene expression in all four cell populations. VEGFCc156s treatment reversed many of these changes in heart and diaphragm LECs but not in cardiomyocytes or cardiac BECs, confirming the specificity of VEGFCc156s effects on the lymphatic vasculature (*Figure 4A–C*, *Figure 4—figure supplement 1F–1H*). A causal reasoning analysis was performed to predict potential upstream regulators for the modulated genes. This analysis predicted that angiotensin II is a causal factor for the gene expression changes in cardiomyocytes from both the angiotensin II-infused and the VEGFCc156s-treated, angiotensin II-infused group, confirming similar effects of angiotensin II on cardiomyocytes at endpoint (*Figure 4—figure supplement 1I*). One common causal factor and related pathway that was activated by VEGFCc156s treatment in LECs from both heart and diaphragm is MYCN, suggesting the activation of lymphatic endothelial cell growth and proliferation by VEGFCc156s (*Figure 4—figure supplement 1J*). Notably, angiotensin II infusion upregulated the expression of multiple genes involved in inflammation, chemotaxis and related pathways in LECs from both heart and diaphragm.

Conversely, VEGFCc156s treatment downregulated the expression of these inflammatory and chemotactic genes (*Figure 4A–C*). Causal reasoning analysis also suggested that toll-like receptor 4 (TLR4) and toll-like receptor 3 (TLR3) signaling pathways were activated in LECs from angiotensin II-infused hearts but were reduced in LECs from VEGFCc156s-treated, angiotensin II-infused hearts, supporting a beneficial effect of VEGFCc156s treatment on angiotensin II infusion-induced inflammatory pathways in LECs (*Figure 4—figure supplement 1J*).

To validate the findings from the RNA-seq analysis, we examined circulating chemokine levels as well as chemokine levels in skin, which is highly vascularized with lymphatic capillaries. Consistent with the RNA-seq analysis, a significant 43% increase in plasma KC/GRO levels was noted in angiotensin II-infused mice compared with control mice, and VEGFCc156s treatment trended to reduce plasma KC/GRO levels in angiotensin II-infused mice (*Figure 4D*). Furthermore, we observed significantly increased levels of several chemokines (37.2% for chemokine interferon γ-induced protein 10 (IP-10), 68% for macrophage-derived chemokine (MDC) and 30% for KC/GRO on average) in skin of angiotensin II-infused mice compared with control mice, while VEGFCc156s treatment significantly reduced the levels or induced a trend toward a reduction in these chemokines (*Figure 4E and F*, *Figure 4—figure supplement 1K*). In addition, angiotensin II infusion significantly increased general macrophage density in heart as well as macrophage density near arteries as assessed by CD68 immunostaining, indicating increased

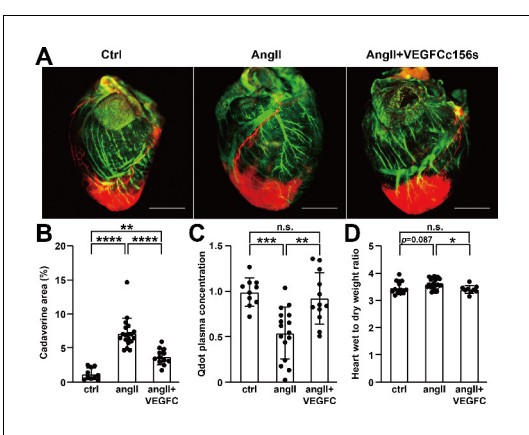

**Figure 3.** VEGFC$_{C156S}$ treatment improved cardiac lymphatic vascular function. (**A**) Representative lymphangiography images of adult mouse hearts. FITC-dextran (green) indicates blood vessels, cadaverine (orange) indicates vascular leak and qdot (red) indicates injection site and cardiac lymphatics. Objective, 1x. Scale bar, 2000 μm. (**B**) Analysis of cadaverine-positive area (normalized to total area) in heart 5 mins after injection. The data were normalized to ctrl group and represented as fold change (*n* = 10–19/group). (**C**) Analysis of qdot plasma concentration 5 mins after injection (*n* = 10–16/group). (**D**) Wet-to-dry weight ratio of hearts (*n* = 9–22/group). Ctrl, control. AngII, angiotensin II. VEGFC, VEGFCc156s. Data are mean ± s.d. One-way *ANOVA* with Bonferroni posthoc was used for statistics. *$p<0.05$, **$p<0.01$, ***$p<0.001$, ****$p<0.0001$, n.s. not significant.

The online version of this article includes the following source data and figure supplement(s) for figure 3:

**Source data 1.** Raw data supporting *Figure 3*.
**Figure supplement 1.** Single-channel lymphangiography images.

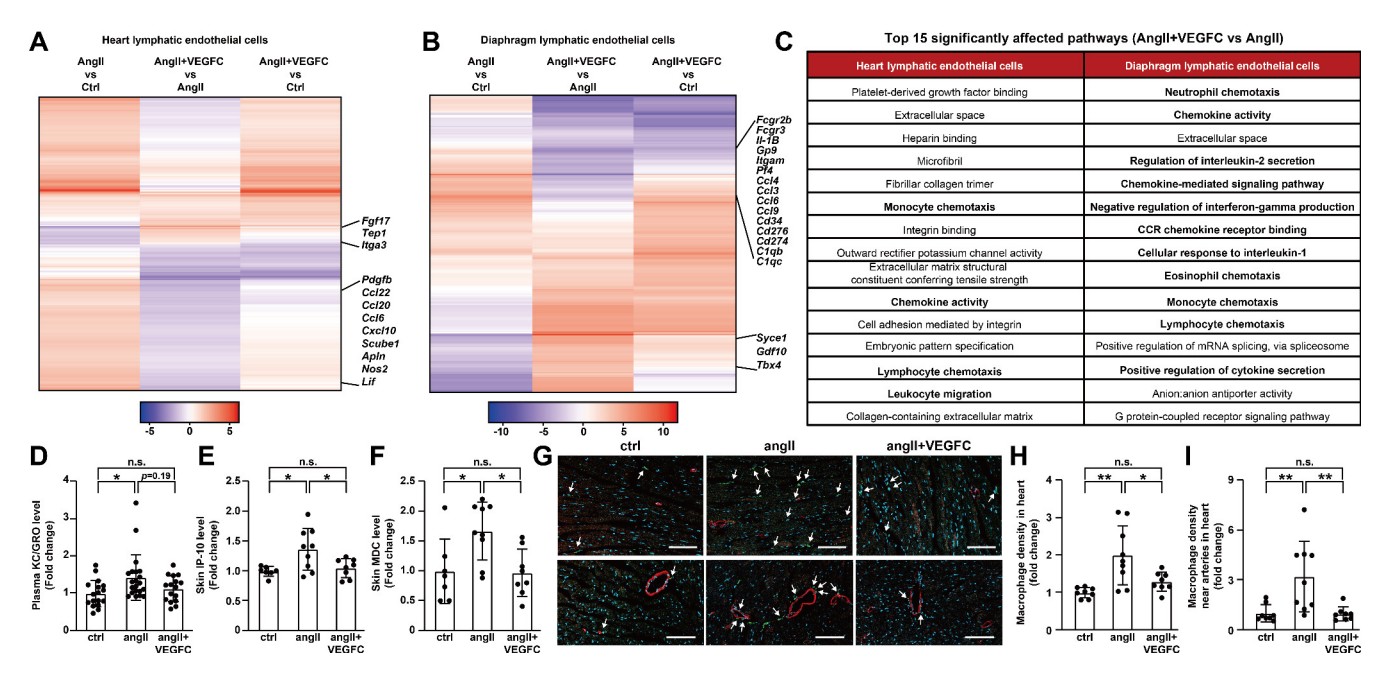

**Figure 4.** VEGFC<sub>C156S</sub> treatment ameliorated angiotensin II-induced inflammatory responses. (A–B) Heat map visualization of all FDR-significant differential expression changes in genes comparing angII and ctrl groups (angII vs ctrl), angII+VEGFC and angII groups (angII+VEGFC vs angII), and angII+VEGFC and ctrl groups (angII+VEGFC vs ctrl) in heart lymphatic endothelial cells (LECs) and diaphragm LECs. The differential expression test-statistics are indicated by the shading in the heatmap and several genes that are significantly regulated by VEGFC treatment are shown. (C) Top 15 affected signaling pathways in angII+VEGFC vs angII comparison for heart LECs and diaphragm LECs from gene ontology (GO) pathway analysis. (D) KC/GRO levels in mouse plasma ($n = 16–21$/group). (E–F) IP-10 and MDC levels in native mouse skin lysates. The values of IP-10 levels were log-transformed, and the values were normalized to the ctrl group and represented as fold change ($n = 7–9$/group). (G) Mouse hearts were fixed, OCT embedded, sectioned, and stained for CD68 (green), smooth muscle actin (red) and DAPI (light blue). Representative images for each group are shown and CD68/DAPI double-positive macrophages are indicated with arrowheads. Scale bar, 100 μm. (H–I) CD68/DAPI double-positive macrophage density and macrophage density near smooth muscle actin-positive arteries (normalized to total area and represented as fold change to ctrl group) in mouse hearts ($n = 8–9$/group). Ctrl, control. AngII, angiotensin II. VEGFC, VEGFCc156s. Data are mean ± s.d. One-way *ANOVA* with Bonferroni posthoc was used for all figures except 4E (One-way *ANOVA* with Tukey's posthoc was used for log-transformed data). \*$p<0.05$, \*\*$p<0.01$, n.s. not significant. The online version of this article includes the following source data and figure supplement(s) for figure 4:

**Source data 1.** Raw data supporting *Figure 4*.

**Figure supplement 1.** Isolation of cardiomyocytes, lymphatic endothelial cells and blood endothelial cells using Prox1-eGFP mice for RNA-seq and additional data analysis.

---

local macrophage content, whereas VEGFCc156s treatment significantly reduced these parameters (*Figure 4G–I*). Altogether, the results suggest that VEGFCc156s treatment ameliorated angiotensin II infusion-induced inflammatory and chemotactic responses potentially by affecting the LEC transcriptome, which may contribute to the efficacy observed in the disease model.

## VEGFCc156s ameliorated angiotensin II-induced hypertension

Immune cells and inflammatory processes can play a role in the pathogenesis of hypertension (*Caillon et al., 2019*), and angiotensin II-induced hypertension promotes cardiac dysfunction. Therefore, the effect of VEGFCc156s treatment on angiotensin II-induced hypertension was examined by conducting a telemetry study as detailed in *Figure 5A*. Telemetry recordings started 3 days before minipump implantation and lasted for 5 weeks after surgery. The time-course analysis of mean blood pressure demonstrated a significant increase in blood pressure and the development of hypertension in angiotensin II-infused mice compared with control mice. Interestingly, VEGFCc156s-treated, angiotensin II-infused mice showed a similar significant increase in both systolic and diastolic blood pressure as those that were treated with angiotensin II alone in the first week. However, starting at week 2, the VEGFCc156s-treated angiotensin II group demonstrated a gradual return in systolic and

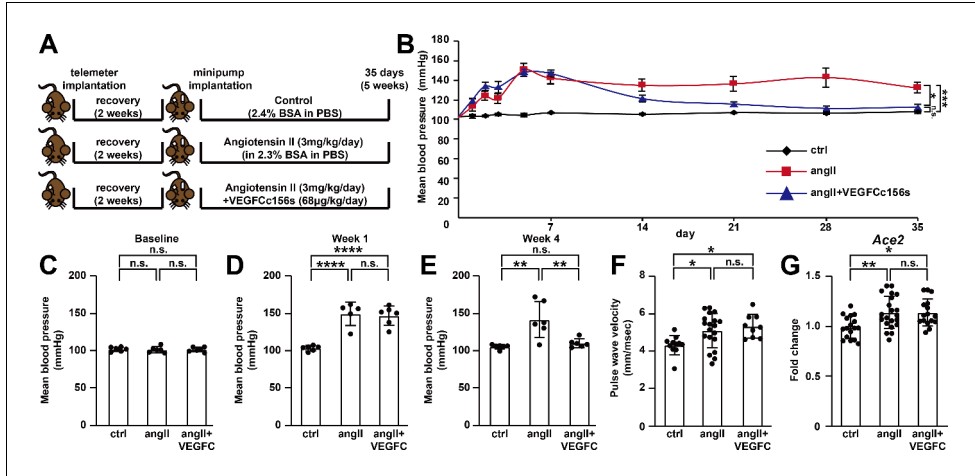

**Figure 5.** VEGFC$_{C156S}$ treatment ameliorated angiotensin II-induced hypertension. (**A**) Experimental design for telemetry study. (**B**) Mean blood pressure of control, angII and angII+VEGFCc156s mice after minipump implantation (a linear mixed model was used for statistics). (**C–E**) Analysis of mean blood pressure at baseline (**C**), week 1 (**D**) and week 4 (**E**) (*n* = 5–6/group). (**F**) Pulse wave velocity assessment (aortic stiffness) 4.5 weeks after minipump implantation (*n* = 9–19/group). (**G**) RNA was extracted from mouse kidneys and qPCR was performed for *Ace2* and normalized to *Hprt*. The data were normalized to the ctrl group and represented as fold change (*n* = 16–21/group). Ctrl, control. AngII, angiotensin II. VEGFC, VEGFCc156s. Data are mean ± s.d. One-way *ANOVA* with Bonferroni posthoc was used for statistics. *$p<0.05$, **$p<0.01$, ***$p<0.001$, ****$p<0.0001$, n.s. not significant.

The online version of this article includes the following source data and figure supplement(s) for figure 5:

**Source data 1.** Raw data supporting *Figure 5*.

**Figure supplement 1.** VEGFC$_{C156S}$ treatment ameliorated angiotensin II-mediated blood pressure increase.

diastolic blood pressure that was comparable to control mice by week 4 (*Figure 5B–E*, *Figure 5— figure supplement 1*). To examine whether VEGFCc156s alleviated angiotensin II infusion-mediated aortic remodeling to reduce blood pressure, pulse wave velocity was assessed to examine aortic stiffness, and both angiotensin II-infused mice and VEGFCc156s-treated, angiotensin II-infused mice showed a significant increase in pulse wave velocity compared with control mice, indicating that VEGFCc156s treatment did not alter angiotensin II infusion-mediated aortic stiffness and remodeling (*Figure 5F*). In addition, the expression of renal *Ace2* mRNA, which encodes the degradation enzyme for angiotensin II, was induced in both angiotensin II-infused mice and VEGFCc156s-treated, angiotensin II-infused mice to a similar extent compared with the kidneys from control mice, indicating a similar compensatory response in kidney to cope with continuous angiotensin II infusion (*Figure 5G*). These results suggest that VEGFCc156s treatment ameliorated angiotensin II-induced hypertension without altering angiotensin II infusion-mediated aortic stiffness and the subsequent renal compensatory response, elucidating another mechanism underlying the beneficial effects of VEGFCc156s on angiotensin II-induced cardiac dysfunction.

## VEGFCc156s improved angiotensin II-induced kidney dysfunction

To validate the effect of VEGFCc156s treatment on angiotensin II infusion-induced hypertension, angiotensin II-induced pathological changes in kidney were investigated. Based on the evaluation of hematoxylin and eosin (H and E) stained sections by a veterinary pathologist, angiotensin II infusion induced vascular hypertrophy in the kidney of both angiotensin II-infused and VEGFCc156s-treated angiotensin II-infused animals, which is consistent with the putative action of angiotensin II as a vasoactive molecule (*Mehta and Griendling, 2007*). However, angiotensin II-infused VEGFCc156s-treated animals showed a reduced incidence of angiotensin II-induced secondary disturbances such as interstitial mononuclear cell infiltrates and tubular basophilia in the kidney compared with angiotensin II-infused animals, indicating that VEGFCc156s treatment partially alleviated angiotensin II infusion-induced kidney injury (*Figure 6—figure supplement 1A and B*). In addition, picrosirius red

(PSR) staining demonstrated increased fibrosis with angiotensin II infusion, and VEGFCc156s treatment significantly reduced the total and perivascular PSR-positive area in kidneys of angiotensin II-infused mice, indicating that VEGFCc156s treatment alleviated angiotensin II-induced kidney fibrosis and injury (*Figure 6—figure supplement 1A, C and D*). Moreover, a significant increase in urine volume was observed in angiotensin II-infused mice compared with control mice over 24 hr, while the angiotensin II-infused, VEGFCc156s-treated mice displayed a significant reduction in urine volume (*Figure 6A*). Urine composition was also evaluated, and consistent with the pathological evaluation, angiotensin II infusion caused kidney damage and significantly increased the amount of non-esterified fatty acids (NEFA), total urinary protein, and the albumin-to-creatinine ratio (ACR) (*Figure 6B–D*). In contrast, VEGFCc156s treatment significantly reduced these parameters compared with animals that had been treated with angiotensin II alone, suggesting that VEGFCc156s treatment alleviated angiotensin II-induced kidney damage. Taken together, these results demonstrate that VEGFCc156s treatment improved angiotensin II-induced kidney dysfunction.

Recent reports indicated a role of renal lymphatics in the pathogenesis of hypertension and demonstrated that renal lymphatics respond to systemic hypertension by expansion (*Balasubbramanian et al., 2020*; *Lopez Gelston et al., 2018*). Therefore, we examined lymphatic endothelial marker expression and lymphatic vascular density in kidney. Angiotensin II infusion significantly increased the mRNA expression levels of lymphatic endothelial markers *Lyve1*, *Pdpn*, *Vegfc*, and *Flt4* in kidney, which is consistent with previous reports (*Balasubbramanian et al., 2020*; *Lopez Gelston et al., 2018*) and represents an endogenous expansion in response to systemic

hypertension. Surprisingly, VEGFCc156s treatment significantly reduced, rather than increased, lymphatic endothelial marker expression in angiotensin II-infused mouse kidneys to levels that were comparable to control mice (*Figure 6E and F*, *Figure 6—figure supplement 1E and F*). Consistent with the gene expression analysis, immunohistochemistry also showed a trend toward increased Lyve1-positive lymphatic vessel density in angiotensin II-infused kidneys, and VEGFCc156s treatment ameliorated this response (*Figure 6—figure supplement 1G and H*). In summary, these results suggest that VEGFCc156s treatment improved angiotensin II-induced kidney dysfunction and ameliorated the response of renal lymphatics to systemic hypertension, supporting the observation that VEGFCc156s ameliorated angiotensin II infusion-induced hypertension.

## VEGFCc156s alleviated angiotensin II-induced cardiac fibrosis and inflammation at 1 week after treatment

To characterize the time-course of VEGFCc156s-mediated protection from angiotensin II-induced cardiac dysfunction and dissect the potential direct effects of VEGFCc156s on the heart from its systemic effect to reduce blood pressure, we examined the effect of VEGFCc156s on angiotensin II-infused animals after 1 week of infusion, when blood pressure was similar between angiotensin II-infused and VEGFCc156s-treated, angiotensin II-infused mice (*Figure 5*). To further characterize the lymphangiogenic response of VEGFCc156s in healthy animals, we added an

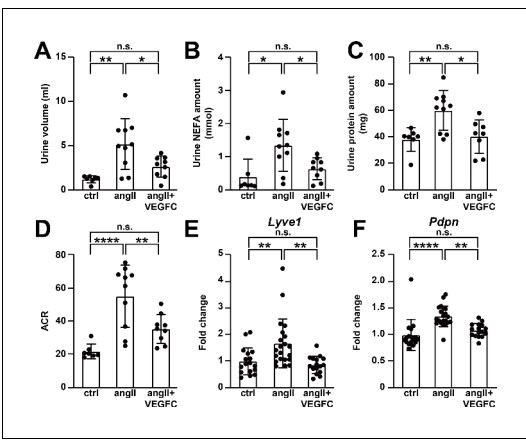

**Figure 6.** VEGFC$_{C156S}$ treatment improved angiotensin II-induced kidney dysfunction. (**A–D**) Urine was collected 4.5 weeks after minipump implantation. Urine volume (**A**), Urine non-esterified fatty acid (NEFA) levels (**B**), total urinary protein (**C**) and albumin to creatinine ratio (ACR) (**D**) are shown (n = 7–10/group). (**E–F**) RNA was extracted from whole kidney, and quantitative RT-PCR was performed for *Lyve1* (**E**) and *Pdpn* (**F**) and normalized to the expression of housekeeping gene *Hprt*. The data were normalized to the control group and represented as fold change (n = 16–21/group). Ctrl, control. AngII, angiotensin II. VEGFC, VEGFCc156s. Data are mean ± s.d. One-way *ANOVA* with Bonferroni posthoc was used for statistics. *$p<0.05$, **$p<0.01$, ****$p<0.0001$, n.s. not significant. The online version of this article includes the following source data and figure supplement(s) for figure 6:

**Source data 1.** Raw data supporting *Figure 6*.
**Figure supplement 1.** Analysis of pathological features, gene expression, and lymphatic vessel density in mouse kidneys.

additional group in which animals were treated with VEGFCc156s alone (*Figure 7A*). As expected, VEGFCc156s infusion increased plasma VEGFC concentration in animals treated with VEGFCc156s alone as well as in animals treated with angiotensin II and VEGFCc156s similarly as was observed in the longer study (*Figure 7—figure supplement 1A*). As a surrogate for blood pressure measurements, we measured hemoglobin levels and calculated hematocrit percentage, which positively correlates with blood pressure (*Atsma et al., 2012*). Significant and comparable increases in hemoglobin levels and hematocrit percentage were observed in the angiotensin II-infused animals compared with the control groups, consistent with an increase in blood pressure in angiotensin II-infused groups (*Figure 7—figure supplement 1B and C*). As an additional control, we examined renal *Ace2* mRNA expression and confirmed its similar induction in both angiotensin II-infused groups compared with the control groups, indicating induction of a compensatory response in the kidney and further validating the action of angiotensin II on both treatment groups at endpoint (*Figure 7—figure supplement 1D*).

Kidney function was examined in these animals by measuring urinary ACR, and no significant change was observed among the groups (*Figure 7—figure supplement 1E*), indicating that the animals had not developed significant kidney dysfunction with a 7-day angiotensin II infusion. We further evaluated kidney expression of lymphatic endothelial marker *Lyve1* as well as lyve1-positive lymphatic vessel density and found no significant change in both parameters among the groups (*Figure 7—figure supplement 1F–1H*), indicating renal lymphangiogenesis was not yet observed with a 7-day infusion of angiotensin II or VEGFCc156s.

To characterize systemic lymphangiogenic responses induced by VEGFCc156s after 7 days of infusion, lymphatic vessel density in skin and heart was examined. We found no significant changes in lyve1-positive lymphatic vessel density in skin at this time point (*Figure 7—figure supplement 1I and J*). However, animals treated with VEGFCc156s alone or in combination with angiotensin II showed a significant increase in podoplanin-positive lymphatic vessel density in heart compared with other groups (*Figure 7—figure supplement 1K and L*).

To investigate whether the lymphangiogenesis response observed in heart in the VEGFCc156s-treated animals altered cardiac function, echocardiography was conducted. While echocardiography demonstrated no significant change in cardiac contractility among the animals, a significant increase in left ventricular mass and ventricular wall thickness in angiotensin II-infused mice was observed, indicating the development of cardiac hypertrophy at 1 week after angiotensin II infusion (*Figure 7B–E*, *Figure 7—figure supplement 1M–1O*). Interestingly, VEGFCc156s treatment did not significantly alleviate angiotensin II-induced cardiac hypertrophy at 1 week after treatment by echocardiography. Similarly, a significant increase in heart weight and a trend towards increased lung weight were observed in angiotensin II-infused animals compared with the control groups, and VEGFCc156s treatment did not alter heart and lung weights in angiotensin II-infused animals at this time point (*Figure 7F and G*).

Despite the lack of alterations in cardiac function with 7 days of VEGFCc156s infusion, a significant increase in *nppb* mRNA expression was observed in angiotensin II-infused hearts, and VEGFCc156s treatment restored *nppb* expression in angiotensin II-infused hearts back to a level that was comparable with the control hearts (*Figure 7H*). Furthermore, a significant increase in the mRNA expression of profibrotic markers (*Col1a1*, *Col1a2*, *Col3a1* and *Fn1*) was observed in angiotensin II-infused hearts compared with control hearts, while animals that had also been treated with VEGFCc156s demonstrated a significant reduction in the expression of these markers (*Figure 7I and J*, *Figure 7—figure supplement 1P and Q*), indicating a potential beneficial effect of VEGFCc156s to reduce angiotensin II infusion-induced cardiac fibrosis. To support this observation, PSR staining was performed in mouse heart sections. Consistent with the gene expression analyses, angiotensin II-infused hearts showed a significant increase in PSR-positive area, indicating the development of cardiac fibrosis, while VEGFCc156s induced a trend toward a reduced PSR-positive area percentage in angiotensin II-infused hearts (*Figure 7K and L*), suggesting that VEGFCc156s ameliorated angiotensin II infusion-induced cardiac fibrosis.

Gene expression analyses of proinflammatory markers revealed that angiotensin II infusion for 7 days significantly increased the mRNA expression of multiple inflammatory markers, cytokines, and chemokines (*Figure 7M*, *Figure 7—figure supplement 2A–2I*). Conversely, VEGFC156s treatment significantly decreased the expression of these proinflammatory markers in angiotensin II-infused hearts, suggesting that VEGFCc156s alleviated angiotensin II-induced cardiac inflammation.

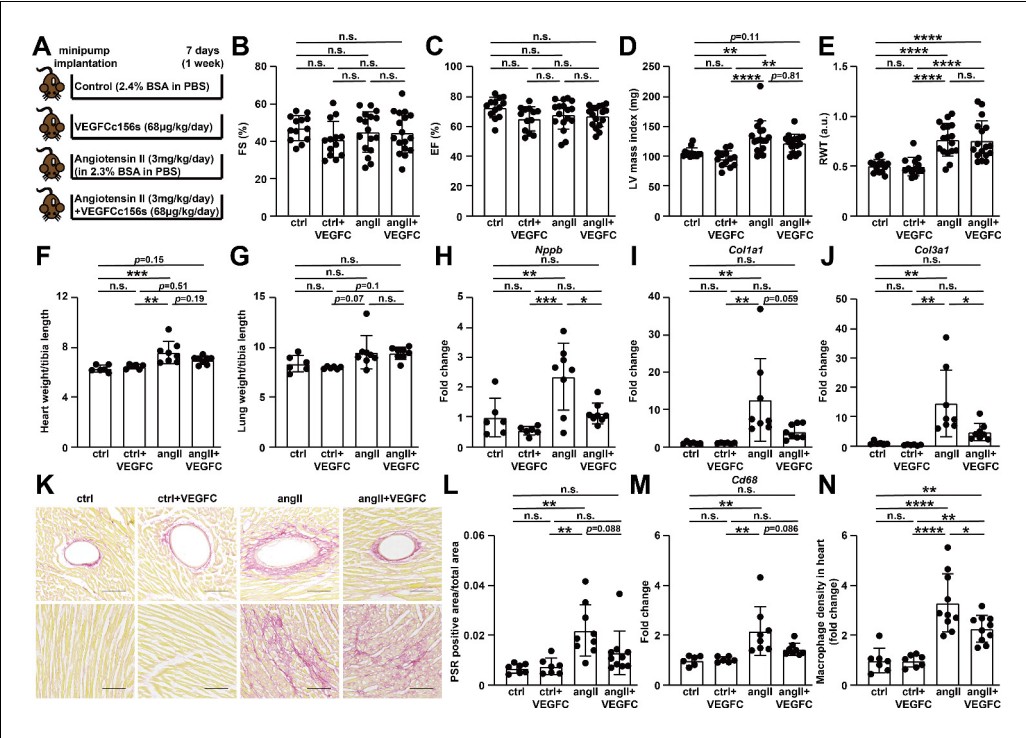

**Figure 7.** VEGFC$_{C156S}$ treatment reduced the gene expression of cardiac dysfunction, fibrosis and inflammatory markers, and alleviated cardiac fibrosis and inflammation at 1 week after treatment. (**A**) Experimental design for the animal study. Angiotensin II (angII) was infused to induce cardiac dysfunction, and VEGFCc156s (VEGFC) was infused as a lymphangiogenic therapy via a subcutaneously-implanted osmotic pump. Bovine serum albumin (BSA) was loaded to the pumps for control (ctrl) and angII groups to balance for the loading of VEGFCc156s. (**B–E**) Echocardiography was conducted at day 6 after minipump implantation. Fractional Shortening (FS) (**B**), Ejection Fraction (EF) (**C**), Left Ventricular (LV) mass index (**D**) and Relative Wall Thickness (RWT) (**E**) are shown ($n$ = 13–17/ group). (**F–G**) Heart weight (**F**) and lung weight (**G**) (normalized to tibia length) of the animals at euthanasia ($n$ = 6– 8/group). (**H–J**) RNA was isolated from the mouse hearts, and quantitative RT-PCR was performed for *Nppb*(H), *Col1a1*(I), *Col3a1*(J), and normalized to the expression of housekeeping gene *Hprt*. The data were normalized to ctrl group and represented as fold change ($n$ = 6–8/group). (**K**) Hearts from the mice described in (**A**) were fixed, sectioned, and stained. Representative Picrosirius red (PSR) stains of mouse heart sections are shown. Scale bar, 100 μm. (**L**) Quantification of PSR-positive area (normalized to total area) on PSR-stained heart sections ($n$ = 7–10/ group). (**M**) Quantitative RT-PCR was performed for *Cd68* and normalized to the expression of housekeeping gene *Hprt*. The data were normalized to ctrl group and represented as fold change ($n$ = 6–8/group). (**N**) CD68/DAPI double-positive macrophage density (normalized to total area and represented as fold change to ctrl group) in mouse hearts ($n$ = 7–10/group). Data are mean ± s.d. One-way *ANOVA* with Bonferroni posthoc was used for statistics. *$p<0.05$, **$p<0.01$, ***$p<0.001$, ****$p<0.0001$, n.s. not significant.

The online version of this article includes the following source data and figure supplement(s) for figure 7:

**Source data 1.** Raw data supporting *Figure 7*.

**Figure supplement 1.** Effect of VEGFC$_{C156S}$ on the lymphatic vessels in heart, kidney and ear skin at 1 week after treatment.

**Figure supplement 2.** VEGFC$_{C156S}$ treatment reduced the gene expression of inflammatory markers in heart and alleviated cardiac inflammation at 1 week after treatment.

Immunohistochemistry was performed to visualize CD68-positive macrophages in heart. A significant increase in CD68-positive macrophage density was observed in angiotensin II-infused hearts compared with control hearts, and VEGFCc156s treatment significantly reduced CD68-positive macrophage density in angiotensin II-infused hearts (*Figure 7N*, *Figure 7—figure supplement 2J*). In summary, VEGFCc156s treatment ameliorated angiotensin II-induced pathological gene expression and angiotensin II-induced cardiac fibrosis and inflammation at 1 week after treatment, a time point at which no divergence in blood pressure is observed.

## Discussion

Recent studies indicated that the lymphatic vasculature, a traditionally-neglected target for therapeutic interventions, plays an important role in the pathogenesis of cardiovascular diseases, and enhancing lymphangiogenesis could be beneficial for acute cardiac injuries (*Henri et al., 2016*; *Houssari et al., 2020*; *Huang et al., 2017*; *Klotz et al., 2015*; *Liu and Oliver, 2019*; *Shimizu et al., 2018*; *Tatin et al., 2017*; *Trincot et al., 2019*; *Vieira et al., 2018*; *Vivien et al., 2019*). However, the role of the lymphatic vasculature in chronic cardiac dysfunction remains to be elucidated. In this study, we aimed to investigate the role of the lymphatic vasculature in the pathogenesis of chronic heart failure and cardiac dysfunction. We found several SNPs near lymphatic marker gene loci that were associated with human cardiovascular or metabolic comorbidities in human GWAS studies, although some of the associations did not meet the genome-wide threshold and some of the loci are close to another gene in addition to the lymphatic endothelial marker (*Table 1*). To validate the findings from the GWAS analysis, we conducted expression profiling using human heart samples and found reduced expression of lymphatic endothelial markers in human chronic heart failure hearts (*Figure 1*), suggesting that the lymphatic vasculature may play a role in the pathogenesis of human chronic heart failure and cardiac dysfunction. Our observations are in line with previously published RNA sequencing analyses of human hearts (*Sweet et al., 2018*; *van Heesch et al., 2019*) and a recent clinical report indicating that serum VEGFC levels are significantly and inversely associated with all-cause and cardiovascular death (*Wada et al., 2018*), supporting a role of lymphatic vasculature in the pathogenesis of cardiovascular diseases. However, previous reports found minimal changes in lymphatic vascular density in left ventricular endomyocardial biopsies from terminal heart failure hearts and healthy donor hearts using immunohistochemistry (*Dashkevich et al., 2009*; *Dashkevich et al., 2010*). Thus, further investigations are required to characterize the changes in lymphatic vasculature in human chronic heart failure pathogenesis.

Using a murine model of cardiac dysfunction, we demonstrate that a chronic cardiac insult such as angiotensin II infusion induced lymphatic vascular dysfunction by reducing lymphatic vascular density (*Figure 2*) and impairing lymphatic vascular transport function in the heart (*Figure 3*). Podoplanin is widely used as a marker for LECs and is preferentially expressed in pre-collectors and collectors of the cardiac lymphatic vasculature (*Henri et al., 2016*). A previous study demonstrated that enhancing lymphangiogenesis by over-expressing VEGFC with adenoviruses in a mouse model of lymphedema induced robust growth of lymphatic capillaries in axillary tissues after 2 weeks but failed to improve lymphatic vascular leakage, while with prolonged VEGFC stimulation (2–6 months), the lymphatic capillaries underwent an intrinsic remodeling and maturation and became functional collecting vessels, leading to reduced tissue edema and restored lymphatic drainage (*Norrmén et al., 2011*; *Tammela et al., 2007*). Moreover, VEGFCc156s can reportedly directly enhance lymphatic pump activity and lymphatic output in rat mesentery collecting lymphatics (*Breslin et al., 2007*), indicating a potential direct beneficial effect of VEGFCc156s on lymphatic vascular function in addition to its role in enhancing lymphangiogenesis and maturation. Here, consistent with these previous reports, we observed that 6 week continuous infusion of VEGFCc156s concomitant with angiotensin II restored collecting vessel function and improved cardiac lymphatic drainage, leading to improved cardiac fluid balance regulation. Myocardial edema is a key clinical feature of cardiac dysfunction and chronic heart failure, and heart function could be significantly compromised with a subtle increase in interstitial fluid volume (*Dongaonkar et al., 2010*). Therefore, the improved cardiac lymphatic vascular function and tissue fluid homeostasis in the VEGFCc156s-treated, angiotensin II-infused animals might contribute to the improved cardiac function and reduced cardiac hypertrophy observed. One area for future investigation is a further characterization of the changes in cardiac lymphatic valves, which play an important role in regulating the lymph flow in collecting vessels. In addition, podoplanin, a marker of lymphatic collecting vessels, is also reportedly expressed in inflammatory cells. Previous reports indicated that a high frequency of podoplanin-expressing cells appeared in the infarcted heart and facilitated local inflammation, and podoplanin neutralization improved cardiac function after myocardial infarction (*Cimini et al., 2017*; *Cimini et al., 2019*; *Quintanilla et al., 2019*). In our study, we assessed podoplanin-expressing lymphatic vessels but not other podoplanin-expressing cells; further analysis of other podoplanin-expressing cells in our disease model would be an area for future investigation.

Previous studies have demonstrated impairments in the lymphatic vasculature in a myocardial infarction model, where the left anterior descending artery is surgically ligated (*Henri et al., 2016*; *Klotz et al., 2015*). Importantly, the lymphatic vessel parallel to this major blood vessel is likely also mechanically impaired by this surgical manipulation, thus testing the hypothesis that mechanical disruption of the lymphatic vasculature can impair heart function, and regrowth using lymphangiogenic therapy is beneficial. Our observations are in line with these previous studies; however, the studies presented herein importantly assess lymphatic dysfunction in the absence of direct injury or surgical manipulation. Likely the impairments in lymphatic vascular function observed in angiotensin II-infused mice could come directly from angiotensin II actions on the endothelium, as angiotensin II can induce endothelial dysfunction (*Gomolak and Didion, 2014*; *Watanabe et al., 2005*) or perhaps indirectly from angiotensin II infusion-induced hypertension, which reportedly alters lymphatic vascular function (*Dongaonkar et al., 2015*).

The lymphatic vasculature also plays an important role in mediating local inflammation (*Brakenhielm and Alitalo, 2019*), and in acute cardiac injury models, resolution of inflammation following lymphangiogenic therapy has been proposed to be an underlying therapeutic mechanism (*Tatin et al., 2017*; *Vieira et al., 2018*). Our RNA-seq analysis indicated that LECs expressed proinflammatory chemokines after angiotensin II stimulation (*Figure 4*), which is consistent with previous reports indicating a role of angiotensin II in initiating inflammatory and chemotactic responses in tissues and vasculature (*Benigni et al., 2010*; *Suzuki et al., 2003*). Interestingly, VEGFCc156s treatment alleviated the inflammatory responses induced by angiotensin II infusion in LECs by reducing the expression of proinflammatory chemokines, which may contribute to the beneficial effects of VEGFCc156s on angiotensin II-induced cardiac dysfunction. Our results are in line with a previous report indicating that VEGFCc156s treatment downregulated the expression of genes involved in immune regulation and inflammation in dermal LECs without stress or stimulation (*Dieterich et al., 2017*), while here we report that VEGFCc156s treatment ameliorated inflammatory responses in LECs under stress (angiotensin II infusion). Moreover, the transcriptional changes in LECs detected by RNA-seq analysis translated to tissue and systemic levels, and we observed elevations in chemokine levels in skin and plasma in angiotensin II-infused mice, which were alleviated by VEGFCc156s treatment. These results in aggregate suggest that lymphatic vessels may play a more active role in mediating local and systemic inflammation than simply providing a route for immune cell clearance, and VEGFCc156s may directly alleviate systemic inflammation through transcriptional effects on LECs.

More interestingly, upon VEGFCc156s treatment, a significant reduction of angiotensin II-induced increases in blood pressure and kidney damage was observed (*Figure 5* and *Figure 6*). Our observations are in line with a previous report indicating that the treatment of VEGFC, a ligand for both VEGFR2 and VEGFR3, also induced a delayed systemic hemodynamic response to reduce blood pressure and attenuated related renal damage in a high-salt induced hypertension model (*Beaini et al., 2019*). Our study used VEGFCc156s, a ligand with specificity for VEGFR3, suggesting that VEGFC-mediated effects are indeed mediated by VEGFR3 activation. Furthermore, these effects are not limited to salt-induced pathology, as VEGFCc156s also ameliorated angiotensin II-induced hypertension and renal damage, suggesting that VEGFC may play a broad role to modulate hypertension. Moreover, in line with the observations using the murine models, a recent clinical study also showed that patients with hypertension have significantly lower serum VEGFC levels compared with healthy individuals (*Chachaj et al., 2018*), supporting a potential role of lymphatic vasculature in the pathogenesis of human hypertension. Hypertension is a major risk factor for heart failure and other cardiovascular diseases (*Dorans et al., 2018*; *Tackling and Borhade, 2019*), and anti-hypertensive therapies significantly reduce cardiovascular events and mortality in patients in clinical studies (*Ettehad et al., 2016*). Therefore, the effect of VEGFCc156s on blood pressure might also contribute significantly to its beneficial effects on angiotensin II-induced cardiac dysfunction.

Consistent with previous reports regarding the role of renal lymphatics in the pathogenesis of hypertension (*Balasubbramanian et al., 2020*; *Lopez Gelston et al., 2018*), we also observed an endogenous expansion of renal lymphatics in response to angiotensin II-induced hypertension (*Figure 6*). The previous reports also demonstrated that enhancing renal lymphatic expansion by over-expressing VEGFD prevented the development of hypertension. Therefore, we initially hypothesized that VEGFCc156s might enhance the endogenous renal lymphangiogenesis response and thus ameliorate angiotensin II infusion-induced hypertension. Surprisingly, no lymphangiogenic response was

observed in the kidney of VEGFCc156s-treated angiotensin II-infused animals at both the 1-week and 6-week time points assessed here (*Figure 6E and F*, *Figure 6—figure supplement 1E–1H* and *Figure 7—figure supplement 1F–1H*). Thus, it is possible that the lymphangiogenesis response we observed in the kidney of angiotensin II-infused mice could be an adaptation to hypertension and was therefore ameliorated in VEGFCc156s-treated, angiotensin II-infused mice at the 6-week time-point due to reduced blood pressure. In the previous reports, the authors used a transgenic line to over-express VEGFD by 800-fold over baseline specifically in kidney and observed significant expansion of renal lymphatics, while our study used VEGFCc156s at a low dose that only increased plasma VEGFC levels by 1.6-fold to 1.9-fold on average. In addition, as a low-molecular weight protein, VEGFCc156s will be quickly filtered through glomeruli followed by adsorptive endocytosis and lysosomal degradation (*Franssen et al., 1994*), resulting in poor local retention and exposure of VEGFCc156s in kidney. Therefore, the discrepancy between our study and previous studies could possibly be explained by differences in local exposure of the lymphangiogenic factors with transgenic overexpression and chronic systemic administration.

Although VEGFCc156s significantly ameliorated angiotensin II-induced hypertension, the underlying molecular mechanisms remain to be explored. We first carefully examined the possibility that VEGFCc156s interfered with angiotensin II-mediated effects, and several lines of evidence in our studies indicate that there were no alterations in angiotensin II-mediated vascular effects after VEGFCc156s treatment, as angiotensin II caused aortic remodeling (*Figure 5F*), induced vascular leakage (*Figure 3B*) and vascular hypertrophy (*Figure 6—figure supplement 1A*) and increased renal *Ace2* expression (*Figure 5G* and *Figure 7—figure supplement 1D*) similarly in both the angiotensin II-infused and the VEGFCc156s-treated, angiotensin II-infused group, indicating that angiotensin II acted as a vasoactive molecule in both groups. In addition, causal reasoning analysis of RNA-seq data also pointed at angiotensin II as a causal factor for the changes observed in cardiomyocytes from both the angiotensin II-infused and the VEGFCc156s-treated, angiotensin II-infused group. Finally, the similar blood pressure elevations detected in the first week in both angiotensin II-infused and VEGFCc156s-treated, angiotensin II-infused groups indicated successful release of angiotensin II from the pumps in both groups (*Figure 5*). Therefore, we suggest that VEGFCc156s did not interfere with angiotensin II effects to alleviate the phenotypes in this disease model. Another possibility is that VEGFCc156s might affect the skin lymphatic vasculature to regulate blood pressure, as skin lymphatics have been implicated as a potential disease-modifying factor in the pathogenesis of salt-sensitive hypertension (*Karlsen et al., 2018*). After 6 weeks of treatment, VEGFCc156s significantly increased skin lymphatic vessel density and diameter in angiotensin II-infused mice (*Figure 2—figure supplement 1O–1R*); these changes may contribute to the reductions observed in blood pressure in these animals. Further investigations are needed to fully characterize the molecular mechanisms underlying the beneficial effect of VEGFCc156s on angiotensin II-induced hypertension.

To dissect the potential direct effect of VEGFCc156s on the heart from its effect on blood pressure, we examined the effects of VEGFCc156s in angiotensin II-infused mice at 1 week after treatment, at which timepoint the blood pressure was similarly high in animals treated with angiotensin II alone or a combination of angiotensin II and VEGFCc156s (*Figure 5*). Although VEGFCc156s treatment enhanced podoplanin-positive lymphatic vessel growth in the heart at this timepoint, it did not alleviate cardiac hypertrophy in angiotensin II-infused mice. More interestingly, VEGFCc156s treatment restored angiotensin II-induced pro-fibrotic and pro-inflammatory markers in angiotensin II-infused hearts in gene expression analyses and immunohistochemistry staining, indicating a beneficial effect of VEGFCc156s in a blood pressure-independent manner (*Figure 7* and *Figure 7—figure supplements 1* and *2*). The lymphatic vasculature has been proposed to play an important role in regulating myocardial fibrosis (*Brakenhielm et al., 2020*) and a previous report indicated that lymphangiogenic factor treatment in a rat model of myocardial infarction improved cardiac function by reducing cardiac fibrosis (*Henri et al., 2016*). Our results are in line with this previous report, as VEGFCc156s also reduced cardiac fibrosis in angiotensin II-infused mice as early as 1 week post-treatment, which may contribute to the efficacy we observed at the 6-week timepoint.

In summary, our study demonstrates that lymphatic vessels are impaired in cardiac dysfunction, and the lymphangiogenic therapy VEGFCc156s prevented cardiac dysfunction by improving lymphatic vascular function, alleviating cardiac fibrosis and inflammation, and ameliorating hypertension. These results provide new insights into the potential effects of a lymphangiogenic therapy on chronic

cardiac diseases and underlying mechanisms and indicate that the lymphatic vasculature could be a therapeutic target for hypertension and hypertensive heart disease.

## Materials and methods

### Animals and treatments

10 to14 weeks old C57/BL6N (#027 purchased from Charles River laboratories) male mice were used for most of the studies, and Prox1-eGFP knock-in male mice were generated using the CRISPR/Cas9 technology and used for in vivo studies at 9–15 weeks old. Bovine serum albumin (A7030, Sigma-Aldrich), human angiotensin II (A9525, Sigma-Aldrich) and recombinant human VEGF-C (Cys156Ser) protein (752-VC, R and D systems) were used as treatments. The sample size was predicted prior to study start with a power analysis. All animal experimental procedures were carried out in accordance with regulations and established guidelines and were reviewed and approved by the Pfizer Institutional Animal Care and Use Committee (AUP # KSQ-2013–00895).

### Minipump implantation

Animals were randomized based on body weight and implanted with a minipump (Alzet 1002 or Alzet 2006 pump) subcutaneously for continuous infusion for 7 days or 42 days. Isoflurane was used as anesthetic in the minipump implantation surgery, and Carprofen in the form of MediGel CPF (Clear $H_2O$) (or Meloxicam SR) and Buprenorphine were used as pain medications for pre-op and post-op care.

### Generation of *Prox1-eGFP* knock-in mouse model using the CRISPR/Cas9 technology

The *Prox1-eGFP* knock-in model was generated by inserting an *IRES-eGFP* cassette into the 3′UTR of the endogenous *Prox1* locus (at 470 bp downstream of the stop codon). For gene editing in one-cell embryos collected from super-ovulated C57BL/6J donors, Cas9 (*Streptococcus pyogenes*) protein (IDT) or mRNA (TriLink), a pair of single guide RNAs (sgRNA) and a donor plasmid were delivered via pronuclear injection. Seed sequences for the sgRNAs were 5′tacccagctgccgagggcat, 5′gtgagctacccagctgccga, and 5′aaccaatgccctcggcagct. The seed sequences were carefully chosen to minimize potential risk of off-target effect. Specifically, these guides lack any potential off-target with off-target score over 1.5 (http://www.benchling.com) across the whole mouse genome. The donor plasmid consists of *IRES-eGFP* flanked by 1.0 kb and 0.8 kb homology arms on the 5′ and 3′ sides, respectively. Founders were identified by genomic PCR with the forward and reverse primers, 5TCCAGGCAACA′GTTCTACAG and 5′TGCACATCAGATTGTCTAAGG, both external to the homology arms. Germline transmission of the targeted allele was achieved by crossing the founders with wild-type C57BL/6J mice.

### Pharmacokinetic study and VEGFC ELISA

Animals were bled via tail snip immediately prior to subcutaneous injection of 0.1 μg/g or 0.3 μg/g human VEGF-C (Cys156Ser) protein dissolved in phosphate buffered saline containing 0.1% BSA. Blood was sampled in EDTA from the animals at 1, 2, 4, 6, and 24 hr after injection via tail snip bleeding. Plasma VEGFC concentration was measured using the human VEGFC Quantikine ELISA kit (DVEC00, R and D systems) or V-PLEX Human VEGFC kit (K151LTD-1, MSD).

### Telemetry study

Telemeters (PA-C10 or HD-X10 from Data Sciences International) were implanted in the animals by cannulating the carotid artery. A small skin incision was made midline over the ventral surface of the neck. The carotid artery was exposed and isolated using blunt dissection. Three lengths of non-absorbable suture were passed underneath the isolated vessel section (proximal occlusion, artery ligature, and distal occlusion). Tension was applied to the proximal and distal sutures to occlude blood flow and elevate the vessel. A small incision was made in the vessel wall with a 25-gauge needle bevel; the catheter was inserted, and tension was released on the proximal occlusion suture. The catheter was advanced beyond the middle and proximal suture approximately 8.0 mm. The middle and proximal sutures were tied around the vessel and catheter. The skin was closed in an interrupted

pattern using 6–0 absorbable monofilament suture. Carprofen in the form of MediGel CPF (Clear H$_2$0) was provided to the animals 2 days prior to surgery and 2 days following surgery, and Buprenorphine was administered post-operatively, and then as needed. The animals recovered for 2 weeks following surgery and telemetry signals were evaluated for acceptable signals. Animals with functional telemeters and stable telemetry signals were used for the following minipump implantation and continuous telemetry recordings. Mean blood pressure, systolic blood pressure, diastolic blood pressure, and heart rate were recorded from the animals at 1 min intervals for 5 weeks. The data for individual day reflect the averaged value over a 24-hr reading period and the weekly data reflect the averaged values over a 3-day reading period. Except necessary post-op care and husbandry needs, any other potential disturbances were minimized. Time-course telemetry recordings were analyzed using a linear mixed model, in which animals are treated as random effects, while time point (i.e. week) and group are treated as fixed effects. Tukey's multiple comparison adjustment was performed for pairwise comparisons in the linear mixed model when the group had a statistically significant impact on the endpoint.

## Echocardiography

Echocardiography was performed in mice under 0.5–1% isoflurane anesthesia using the Vevo 2100 or Vevo 3100 Imaging System (VisualSonics Inc, Toronto, Canada). B-mode images from long axis and M-mode images from short axis were obtained by one trained individual (T.S. or X.C.), and data were analyzed blindly by another trained individual (L.S.) using Vevo Lab. Pulse wave velocity doppler was conducted using a doppler flow velocity system (Indus Instruments). Simultaneous doppler velocity spectrograms were obtained from the aortic arch and the abdominal aorta using two probes, and the recordings were analyzed blindly by L.S. following manufacturer's technical notes. Pulse wave velocity was calculated by dividing the distance between the probes over the time measurement.

## Tissue sampling

Animals were subjected to isoflurane anesthesia, and tissues (heart, lung, chest skin, and kidney) and blood were collected from the animals. Hearts and lungs were weighed, and hearts, chest skins, and kidneys were snap-frozen in liquid nitrogen and stored in −80℃. The tibia was isolated from the animals, and its length was measured with a caliper. Hemoglobin level and hematocrit percentage were measured using an AimStrip Hb Hemoglobin test system (Germaine Laboratories) using whole blood in EDTA. The blood was next centrifuged, and plasma was isolated and stored at −80℃ until being used for plasma VEGFC concentration measurements. For histology, animals were perfused transcardially with PBS containing heparin (20 U/ml), vasodilators (adenosine 1 g/L and papaverine 4 mg/L) and 25 mM potassium chloride (to arrest the heart in diastole) followed by 10% neutral-buffered formalin perfusion. The heart, ear, diaphragm, and kidney were next collected and immersed in 10% neutral-buffered formalin at 4℃ for 24–48 hr.

## Gravimetry

Wet heart weights were measured immediately after euthanasia, and hearts were dried in a 37℃ incubator for 5 days. Dry weights were measured after desiccation. The wet-to-dry weight ratio was calculated by dividing the wet weight over the dry weight for each animal.

## Gene and protein expression analysis

Snap-frozen tissues were smashed, mixed and distributed into small portions for RNA and protein isolation. RNA was isolated from the tissues using Trizol reagent (Thermofisher) and RNeasy mini kits (Qiagen). The RNA was either directly used for qPCR with TaqMan RNA to Ct 1-step kit (ThermoFisher) or cDNA was synthesized from the RNA using high capacity cDNA reverse transcription kit (ThermoFisher) and used for qPCR. qPCR was conducted using TaqMan probes (*Nppb*, Mm01255770_g1; *Pdpn*, Mm01348912_g1; *Lyve1*, Mm00475056_m1; *Flt4*, Mm01292604_m1; *Vegfc*, Mm00437310_m1; *Col1a1*, Mm00801666; *Col1a2*, Mm00483888; *Col3a1*, Mm00802300_m1; *Fn1*, Mm01256744; *Cd68*, Mm03047343_m1; *Cxcl10*, Mm00445235_m1; *Ptprc*, Mm01293577_m1; *Itgax*, Mm00498701_m1; *Itgam*, Mm00434455_m1; *Il1b*, Mm00434228_m1; *Il6*, Mm00446190_m1; *Ccl6*, Mm01302419_m1; *Ccl4*, Mm00443111_m1; *Ccl3*, Mm00441259_g1; *Ace2*, Mm01159006_m1; *Hprt*,

Mm03024075_m1), TaqMan gene expression master mix (ThermoFisher) and QuantStudio 7 Flex Real-Time PCR System. The expression of every gene was normalized to the expression of housekeeping gene *Hprt* and then normalized to control group to be presented as fold change in the corresponding figures.

Podoplanin antibody (AF3244, R and D systems, 1:500), Akt antibody (#9272, cell signaling, 1:1000), p-Akt (Ser473) antibody (#9271, cell signaling, 1:1000), p44/42 MAPK (Erk 1/2) antibody (#9102, cell signaling, 1:2000), phsopho-p44/42 MAPK (Erk1/2) (Thr202/Tyr204) antibody (#4370, cell signaling, 1:2000), and GAPDH antibody (#5174, cell signaling, 1:5000) were used for western blots. For western blots, tissues were lysed using cell lysis buffer (cell signaling) containing Halt protease and phosphatase inhibitor cocktail and EDTA. Protein concentrations were measured using Pierce BCA protein assay kit and samples were denatured using NuPAGE LDS sample buffer and reducing agent at 95°C for 5 min. A total of 20–40 µg of proteins were loaded to each well of NuPAGE 4–12% Bis-Tris Midi gels and ran in MOPS buffer with Xcell4 Surelock midi gel electrophoresis system. The gels were transferred to PVDF membranes using an iblot2 gel transfer device and blocked with 5% milk (cell signaling) for 1 hr under room temperature followed by primary antibody incubation at 4°C overnight. The next day, membranes were washed and incubated with corresponding HRP-conjugated secondary antibody for 1 hr under room temperature. Then the membranes were washed, developed using SuperSignal west Femto maximal sensitivity substrate and imaged with Amersham Imager 600 (GE Healthcare). All reagents were from ThermoFisher unless otherwise stated. Densitometric quantification was conducted using Image J.

## Histology and immunohistochemistry

Formalin fixed hearts, ears, and kidneys were dehydrated and used for either OCT embedding or paraffin embedding. One paraffin—embedded cross-section (5 µm thickness) through the hilus of the kidney from each animal was stained with H and E using an automatic slide stainer (Leica) and microscopic evaluation was performed by a veterinary pathologist. Cryo-sections (10 µm thickness) through the hilus of the kidney were generated for immunohistochemistry. Cryo-section (10 µm thickness) of OCT embedded hearts at different depths spanning the whole ventricle were generated for immunohistochemistry. Cryo-section (10 µm thickness) of OCT embedded ears at different depths spanning the whole ear were generated for immunohistochemistry. Picrosirius red (PSR) stain were conducted using mouse heart or kidney frozen sections following an optimized protocol using Bouin's solution (FXBOULT, American Mastertech), 1% phosphomolybdic acid (26693–08, EMS), 0.1% picrosirius red solution (F-357–2, Rowley Biochemical) and 1% acetic acid (100–1, Ricca). Lyve1 (abcam, ab14917, 1:500), podoplanin (abcam, ab11936, 1:250), CD68 (ThermoFisher, MA5-16674, 1:300), smooth muscle actin (abcam, ab7817,1:100) and GFP (abcam, ab13970, 1:100) immunostaining was performed in 3% goat serum (ThermoFisher) and 2% BSA in 0.3% Triton-X (ThermoFisher) at 4°C overnight, followed by fluorescent secondary antibodies (Invitrogen). Autofluorescence were quenched using trueview autofluorescence quenching kit and sections were mounted in VECTA-SHIELD Vibrance Antifade Mounting Medium with DAPI (vector laboratories). For the whole mount stain of diaphragm, the whole diaphragm of Prox1-eGFP mice was used for staining following the immunostaining protocol for cryo-sections without the autofluorescence quenching step. For the whole mount stain of ear skins, the ear skins were isolated and cleaned, and stained for Lyve1 following the immunostaining protocol for cryo-sections without the autofluorescence quenching step. The whole-mount-stained samples were imaged using a confocal microscope (ZEISS) with 20x objective. WGA stain was conducted using Alexa 594-conjugated WGA (W11262, ThermoFisher) following manufacturer's protocol with one modification that sections were incubated with WGA overnight at 4°C instead of 10 mins under room temperature. The WGA-stained sections were washed after WGA incubation and mounted with Prolong Gold mounting medium with DAPI (P36935, Thermofisher). Stained slides were scanned using a ZEISS Axio Scan Slide Scanner with 20x objective and whole section images were analyzed using Visiopharm and custom-designed apps. For the quantification of CD68-positive macrophages, three to six sections spanning the ventricle were used, and macrophages were defined as CD68 and DAPI double-positive cells. Macrophage density was calculated by dividing the count of macrophages to the total area for each section. The density of macrophages near arteries was defined as the total count of macrophages within 20 µm distance from any smooth muscle actin-positive artery normalized to the total area for each section. The data for macrophage density was normalized to the average value of ctrl group and represented as fold change. For the

quantification of podoplanin-positive lymphatic vessel density and area, 6 to 12 sections spanning the ventricle were used. A size filter of 500 μm$^2$ was used to define the vessels and reduce the noise from podoplanin-positive cells. The count of podoplanin-positive lymphatic vessels was normalized to the total area for each section and defined as the density of podoplanin-positive lymphatic vessels. The average lumen area was calculated by dividing the lumen area of the podoplanin-positive lymphatic vessels by the density of podoplanin-positive lymphatic vessels in each section. A linear mixed model with animals treated as random effects and group and sections treated as fixed effects was used for the statistical analysis for the quantification of podoplanin-positive lymphatic vessel density and lumen area. Tukey's multiple comparison adjustment was used for pairwise comparisons in the linear mixed model. The data for podoplanin-positive lymphatic vessel density was normalized to the average value of ctrl group and represented as fold change. The analysis for the quantification of cardiomyocyte size and variance coefficient was conducted following previously published methods and SOPs for muscle fiber size and size variability quantification (*Briguet et al., 2004*; *Pertl et al., 2013*). The variance coefficient is defined as the ratio of the standard deviation of cardiomyocyte size to the mean value of cardiomyocyte size and represents the variability of cell size among the quantified population. A linear mixed model with defined regions of interest (ROIs) treated as random effects and group treated as fixed effects was used for the statistical analysis for variance coefficient and Tukey's multiple comparison adjustment was used for pairwise comparisons in the model. For the quantification of lyve1-positive lymphatic vessel density in kidney and ear skin, the lyve1-positive lymphatic vessel count was normalized to the total area for each section. A linear mixed model with defined regions of interest (ROI) treated as random effects and group treated as fixed effects was used for the statistical analysis of lyve1-positive lymphatic vessel density. Tukey's multiple comparison adjustment was used for pairwise comparisons in the linear mixed model. For the quantification of PSR stains, three sections spanning the ventricle were used for staining and the images were analyzed using visiopharm. The total PSR-positive area and perivascular PSR-positive area were normalized to total area of the section. For the quantification of lyve1-positive lymphatic vessel diameter in ear skin, the analysis was conducted blindly by L.S. using Image J. Visiopharm-based image analyses were conducted blindly by J.P. All the representative images were taken using a confocal microscope (ZEISS) with 20x or 40x objective.

## Lymphangiography

Lymphangiography was performed in anesthetized mice 5–6 weeks after minipump implantation following a previously reported method with modifications (*Henri et al., 2016*). The animal first received a dose of FITC-Dextran (30 mg/ml, FD500S, MW: 500 kDa, Sigma-Aldrich) mixed with alexa Fluor 555 Cadaverine (6 μg/g, A30677, ThermoFisher) and heparin (20 U/ml) via retro-orbital (R.O.) injection under isoflurane. Next, the mouse was further anesthetized and placed under a surgical plane of anesthesia using isoflurane with intubation through tracheotomy. Ten μl fluorescent quantum dots (Q21021MP, ThermoFisher) were injected intramyocardially into the apex of the heart using a Hamilton neuros micro-injection syringe (Hamilton 6546006). Five min after injection, blood was sampled from the animal in EDTA, and the heart was extracted from the animal, arrested in diastole with saturated potassium chloride and fixed with formalin for imaging. The plasma qdot signal (emission at 655) in 50 μl plasma was measured using a fluorescent plate reader. Whole heart was imaged using a LSM 880 Confocal microscope (ZEISS) with a 1x objective, and HDR imaging was used to enhance image contrast. The same intensity and imaging settings were used for acquiring the cadaverine images for all the animals. The results in *Figure 3* were acquired from two rounds of independent in vivo studies. The cadaverine-positive areas were quantified from at least four images taken from four different sides of each heart and normalized to the total area of the heart using image J. The values were normalized to the average value of the control group for presentation in *Figure 3B*. The qdot plasma concentrations of each group were normalized to the average qdot plasma concentration of the control group for each round of studies and then combined for illustration in *Figure 3C*.

## Cell isolation using Prox1-eGFP mice and low input RNA-seq

Prox1-eGFP mice were used for cell isolation at 5–6 weeks after minipump implantation for preparing RNA-seq samples and wildtype non-carrier mice were used at similar age for preparing single

stain controls. The animals were euthanized using carbon dioxide, and heart and diaphragm were isolated from the animals and transferred into a storage solution containing 2.5% BSA (A7030, Sigma-Aldrich) in HBSS (14025092, Thermo Fisher). The tissues were then cut into tiny pieces and digested with a digestion enzyme mixture with Collegenase A (11088793001, Roche), Collegenase from *Clostridium histolyticum* (C7657, Sigma-Aldrich), Dnase I (DN25, Sigma-Aldrich), Hydraluronidase (HX0514-1, Sigma-Aldrich) and 2.5% BSA in HBSS for 45 min in 37℃ waterbath with shaking. The tissue pieces were next broken up further with pipetting and the enzymes were neutralized with DMEM/F12 medium (10565042, Thermo Fisher) containing 10% FBS (26140087, Thermo Fisher). The solutions with digested tissues were centrifuged at 400 g for 3 min, and cardiomyocytes were pelleted from digested heart tissues. The cardiomyocytes were next washed with PBS once and lysed for RNA with RLT cell lysis buffer. The supernatant of each solution was isolated and filtered through 40-µm cell strainer. The solutions were then centrifuged at 1500 g for 5 min to pellet the cells and the cells were resuspended in red blood cell lysis buffer (11814389001, Roche) to lyse red blood cells. Next, the cells were pelleted and washed with FACS buffer (1% BSA in PBS) twice and cell density was counted. 0.2 million cells were aliquoted either from wild-type animal tissues or Prox1-eGFP tissues for single stain controls. Mouse Fc block (553142, BD) were added to each tube to inhibit non-specific binding and the samples were stained for CD45.2 (560696, BD, 1:100), or/and CD31 (561814, BD, 1:100) for 30 min on ice in the dark. The cells were then washed with FACS buffer twice and resuspended in sort buffer (10 mM HEPES (15630080, Thermo Fisher), 2 mM EDTA (AM9912, Thermo Fisher) and 0.5% BSA in PBS) at 20 million/ml for live cell sorting. DAPI (D1306, Thermo Fisher, 1:5000) was added to the cells before sorting to stain the dead cells and the cells were sorted using SONY SH800 cell sorter with 100 µm chip. Blood endothelial cells were defined as DAPI negative, CD45.2 negative, CD31 positive, and eGFP negative cells and lymphatic endothelial cells were defined as DAPI negative, CD45.2 negative, CD31 positive, and eGFP positive cells. The cells were sorted into RLT lysis buffer at 4℃, and total RNA was extracted from each isolated cell population using the RNeasy MinElute cleanup kit (74204, Qiagen) following manufacturer's protocol. Extracted total RNA was stored at −80℃ freezer until the library preparation step. For low-input RNA-seq, total RNA concentration for each sample was measured using high sensitivity plates on a Lunatic nanodrop instrument (Unchained Labs) before proceeding to library prep. Of total RNA, 2 ng were aliquoted from each sample from which mRNA was isolated and converted into cDNA using the Smart-Seq ultra low input RNA kit (634890, Takara) following manufacturer's protocol. The libraries were next prepared from converted cDNA using Nextera DNA XT Library Prep Kit (FC-131–1096, Illumina) following manufacturer's protocol. Library concentrations were quantified using Qubit 1X dsDNA HS Assay kit (Q33231, Thermo Fisher) and the quality of the libraries was examined using Agilent High Sensitivity D1000 Screentape System (5067–5584 and 5067–5585, Agilent) on an Agilent 4200 Tapestation. The libraries from cardiomyocytes were pooled together and sequenced using NextSeq 500/550 High Output Kit (20024907, Illumina). The libraries from endothelial cells were randomized, divided into two sets and pooled together for two rounds of sequencing using NextSeq 500/550 High Output Kits (20024907, Illumina).

## RNA-seq analysis

Sequencing reads were de-multiplexed using the picard tool bcl2fastq, and quality control was performed on the fastq files using FastQC. The reads were then aligned to the mouse reference genome (GRCm38) using the STAR aligner (v.2.5.3a) with quantMode specified for Transcriptome SAM, out SAM type specified as BAM, out Filter Mismatch Nover Lmax as 0.05, align SJDBoverhang Min as 1, out Filter Score Min Over Lread as 0.9, out Filter Match Nmin Over Lread as 0.9, and align Intron Max as 1000000. Gene counts for each sample were generated from the BAM alignment files using Salmon (v.0.12.0) in alignment-aware mode, based on gene/isoform features annotated by Ensembl release 95. Using the log2 normalized gene counts, we performed principal components analysis across samples within each of the four different cell types to identify outlier samples and removed samples that were greater than three standard deviations from the mean value of the first two principal components. For differential expression analysis, all downstream differential expression analysis and functional enrichment analyses were performed in R (v.3.5.3). We performed differential expression analysis within each of the four different cell types among treatment groups using DESeq2, adjusting for age of the animals, and sequencing run batch for the endothelial cell experiments, as well as surrogate technical variables detected using sva analysis. We additionally

performed differential expression analysis across all three endothelial cell groups to identify potential interactions with treatment using the limma-voom framework. To identify pathways and sets of genes enriched in our significantly differentially expressed results, we performed functional enrichment analysis using the topGO enrichment analysis with the weight01 method. Finally, in order to identify potential upstream regulators of the observed expression changes from our results, we performed causal reasoning analysis based on a previously reported approach (*Chindelevitch et al., 2012*). In order to generate the network to identify the upstream regulators, we used proprietary network data compiled from several different sources, but including information on proteins, mRNAs, compounds, and biological processes.

## Plasma and skin chemokine measurement

Chemokine levels in plasma and native skin lysates were examined using MSD U-plex chemokine combo 1 (Ms) kit (K15321K-1, MSD) following manufacturer's protocol. 20 μg of native skin lysate was loaded to each well. The data was analyzed using MSD discovery workbench software (v4.0).

## Urine collection and biochemistry

Animals were transferred into metabolic cages for urine collection for 24 hr and returned to regular cages after collection. Non-esterified fatty acid and creatinine in urine were measured by Siemens clinical analyzer (ADVIA Chemistry XPT system). Albumin was measured using BCP albumin assay kit (Sigma-Aldrich). Total protein amount was measured using Pierce BCA protein assay kit (ThermoFisher).

## Gene expression analysis of human hearts

Human heart tissues were commercially purchased and three commercially available healthy human heart RNA samples (from Ambion, Takara and Origene) were also purchased and used for the analysis. RNA was extracted from human tissues using Qiazol reagent and RNeasy plus universal kit (Qiagen), and cDNA was synthesized using high-capacity cDNA reverse transcription kit (ThermoFisher). qPCR was conducted using TaqMan probes (*LYVE1*, Hs00272659_m1; *PDPN*, Hs00366766_m1; *FLT4*, Hs01047677_m1; *VEGFC*, Hs01099203_m1; *HPRT1*, Hs02800695_m1), TaqMan gene expression master mix (ThermoFisher) and QuantStudio 7 Flex Real-Time PCR System. The expression of every gene was normalized to the expression of housekeeping gene *HPRT1* and then normalized to the healthy heart group to be presented as fold change in the corresponding figures.

## GWAS result evaluation

GWAS summary statistics were aggregated from publicly available resources, including the GWAS catalog (https://www.ebi.ac.uk/gwas/). Locus-wide variant lookups for associations to cardiometabolic-related traits were performed for genes involved in lymphatic-related pathways, including *VEGFC*, *LYVE1* and *FLT4*. Methods for each GWAS analyses can be found at each reference.

## Statistics

The statistical analysis used for every figure has been illustrated in the corresponding figure legends. Independent two-tailed Student's *t*-tests and one-way *ANOVA* with Bonferroni-adjusted posthoc test for multiple pairwise comparisons were performed using Graphpad Prism 7/8 software. F tests were conducted for comparing variances for the Student's *t*-tests, and Welch's correction was used for the Student's *t*-test when the F test result was significant. Statistical analysis using a linear mixed model and the statistical analysis for log-transformed data were conducted using R statistical software (version 3.6.1). All the fitted models meet the model assumptions. All the samples used in the study were biological repeats, not technical repeats. No samples were excluded from the analysis unless it was determined to be a significant outlier by the Grubbs' test.

## Acknowledgements

We thank Dr. Kendra Bence for helpful discussions and suggestions for the project. We thank Rob Webster and John Litchfield for their help with the pharmacokinetic analysis for the studies. We thank Donald Bennett for helpful advice on biostatistics. We thank Matthew Peloquin and Dr.

Federico Damilano for their helpful advice on experiments. We thank the genetically engineered murine model (GEMM) facility at Pfizer Inc for their support in transgenic animal model generation. The project is supported by funding from Pfizer, Inc

## Additional information

### Competing interests

LouJin Song, Xian Chen, Terri A Swanson, Brianna LaViolette, Jincheng Pang: is an employee at Pfizer inc. Teresa Cunio: was an employee at Pfizer inc and is currently an employee at Acceleron Pharma. Michael W Nagle: was an employee at Pfizer Inc and is currently an employee at Eisai Inc. Shoh Asano, Katherine Hales, Arun Shipstone, Hanna Sobon, Sabra D Al-Harthy, Youngwook Ahn, Steven Kreuser, Andrew Robertson, Casey Ritenour, Frank Voigt, Magalie Boucher, Furong Sun, Rachel J Roth Flach: is an employee at Pfizer Inc. William C Sessa: consultant for Pfizer.

### Funding

| Funder | Author |
| --- | --- |
| Pfizer | Rachel J Roth Flach |

The funders had no role in study design, data collection and interpretation, or the decision to submit the work for publication.

### Author contributions

LouJin Song, Conceptualization, Data curation, Formal analysis, Investigation, Visualization, Writing - original draft, Writing - review and editing; Xian Chen, Terri A Swanson, Brianna LaViolette, Teresa Cunio, Shoh Asano, Katherine Hales, Sabra D Al-Harthy, Steven Kreuser, Andrew Robertson, Casey Ritenour, Frank Voigt, Magalie Boucher, Investigation; Jincheng Pang, Formal analysis, Methodology; Michael W Nagle, Formal analysis, Methodology, Writing - review and editing; Arun Shipstone, Hanna Sobon, Youngwook Ahn, Investigation, Methodology; Furong Sun, Formal analysis; William C Sessa, Supervision, Visualization, Writing - review and editing; Rachel J Roth Flach, Conceptualization, Supervision, Funding acquisition, Visualization, Project administration, Writing - review and editing

### Author ORCIDs

LouJin Song (iD) https://orcid.org/0000-0002-1646-8121
William C Sessa (iD) https://orcid.org/0000-0001-5759-1938
Rachel J Roth Flach (iD) https://orcid.org/0000-0003-2754-828X

### Ethics

Animal experimentation: The ethics statement has been included in the method section of the manuscript: "All animal experimental procedures were carried out in accordance with regulations and established guidelines and were reviewed and approved by the Pfizer Institutional Animal Care and Use Committee (AUP # KSQ-2013-00895)".

### Decision letter and Author response

Decision letter https://doi.org/10.7554/eLife.58376.sa1
Author response https://doi.org/10.7554/eLife.58376.sa2

## Additional files

### Supplementary files

• Transparent reporting form

## Data availability

RNA-Seq data has been deposited in GEO under accession code GSE150041. All other data generated during the study are included in the manuscript and supporting files. Source data has been provided for Figure 1-7.

The following dataset was generated:

| Author(s) | Year | Dataset title | Dataset URL | Database and Identifier |
|---|---|---|---|---|
| Song L, Shipstone A, Nagle MW | 2020 | RNA sequencing study using Prox1-eGFP mice to investigate the role of lymphatic vasculature in chronic cardiac dysfunction | https://www.ncbi.nlm.nih.gov/geo/query/acc.cgi?acc=GSE150041 | NCBI Gene Expression Omnibus, GSE150041 |

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

# Appendix 1

**Appendix 1—key resources table**

| Reagent type (species) or resource | Designation | Source or reference | Identifiers | Additional information |
|---|---|---|---|---|
| Mice, male | C67/BL6N | Charles River laboratories | Catalog: #027 | |
| Mice, male | Prox1-eGFP KI | This paper | | eGFP gene was inserted at the end of endogenous *Prox1* gene to generate this indicator line. The spe rms of the animals are cryopreserved in Charles River Laboratories. |
| Peptide, recombinant protein | Bovine serum albumin | Sigma-Aldrich | A7030 | |
| Peptide, recombinant protein | human angiotensin II | Sigma-Aldrich | A9525 | |
| Peptide, recombinant protein | recombinant human VEGF-C (Cys156Ser) protein VEGFC | R and D systems | 752-VC | |
| Other | minipump | Alzet | Alzet 1002 or Alzet 2006 | |
| Sequence-based reagent | Prox1-eGFP genotyping forward primer | This paper | PCR primer | The primer sequence is 5'TCCAGGCAACAGTTCTACAG. It can be ordered from Integrated DNA Technologies using the sequence. |
| Sequenced-based reagent | Prox1-eGFP genotyping reverse primer | This paper | PCR primer | The primer sequence is 5'TGCACATCAGATTGTCTAAGG. It can be ordered from Integrated DNA Technologies using the sequence. |
| Commercial assay or kit | human VEGFC Quantikine ELISA kit | R and D systems | DVEC00 | |
| Commercial assay or kit | V-PLEX Human VEGFC kit | MSD | K151LTD-1 | |
| Other | Telemeters | Data Sciences International | PA-C10 or HD-X10 | |
| Other | Vevo2100 or Vevo3100 | VisualSonics Inc | | |
| Other | doppler flow velocity system | Indus Instruments | | |
| Other | AimStrip Hb Hemoglobin test system | Germaine Laboratories | | |
| Other | Trizol reagent | ThermoFisher | 15596026 | |

*Continued on next page*

*Appendix 1—key resources table continued*

| Reagent type (species) or resource | Designation | Source or reference | Identifiers | Additional information |
|---|---|---|---|---|
| Commercial assay or kit | RNeasy mini kits | Qiagen | 74104 | |
| Commercial assay or kit | TaqMan RNA to Ct 1-step kit | Thermofisher | 4392938 | |
| Commercial assay or kit | high capacity cDNA reverse transcription kit | Thermofisher | 4368814 | |
| Other | TaqMan gene expression master mix | Thermofisher | 4369016 | |
| Other | Tapman probes for mouse | Thermofisher | *Nppb*, Mm01255770_g1; *Pdpn*, Mm01348912_g1; *Lyve1*, Mm00475056_m1; *Flt4*, Mm01292604_m1; *Vegfc*, Mm00437310_m1; *Col1a1*, Mm00801666; *Col1a2*, Mm00483888; *Col3a1*, Mm00802300_m1; *Fn1*, Mm01256744; *Cd68*, Mm03047343_m1; *Cxcl10*, Mm00445235_m1; *Ptprc*, Mm01293577_m1; *Itgax*, Mm00498701_m1; *Itgam*, Mm00434455_m1; *Il1b*, Mm00434228_m1; *Il6*, Mm00446190_m1; *Ccl6*, Mm01302419_m1; *Ccl4*, Mm00443111_m1; *Ccl3*, Mm00441259_g1; *Ace2*, Mm01159006_m1; *Hprt*, Mm03024075_m1 | |
| Other | QuantStudio 7 Flex Real-Time PCR System | ThermoFisher | | |
| Antibody | Anti-mouse Podoplanin antibody (Goat polyclonal) | R and D systems | AF3244 | 1:500 dilution for Western blot |

*Continued on next page*

*Appendix 1—key resources table continued*

| Reagent type (species) or resource | Designation | Source or reference | Identifiers | Additional information |
|---|---|---|---|---|
| Antibody | anti-Akt antibody (Rabbit polyclonal) | Cell signaling | #9272 | 1:1000 dilution for Western blot |
| Antibody | anti- p-Akt (Ser473) antibody (Rabbit polyclonal) | Cell signaling | #9271 | 1:1000 dilution for Western blot |
| Antibody | p44/42 MAPK (Erk 1/2) antibody (Rabbit polyclonal) | Cell signaling | #9102 | 1:2000 dilution for Western blot |
| Antibody | phsopho-p44/42 MAPK (Erk1/2) (Thr202/Tyr204) antibody (Rabbit monoclonal) | Cell signaling | #4370 | 1:2000 dilution for Western blot |
| Antibody | GAPDH antibody (Rabbit monoclonal) | Cell signaling | #5174 | 1:5000 dilution for Western blot |
| Antibody | Lyve1 antibody (Rabbit polyclonal) | Abcam | ab14917 | 1:500 dilution for IF |
| Antibody | podoplanin antibody (Hamster monoclonal) | Abcam | ab11936 | 1:250 dilution for IF |
| Antibody | CD68 antibody (Rat monoclonal) | ThermoFisher | MA5-16674 | 1:300 dilution for IF |
| Antibody | smooth muscle actin antibody (Mouse monoclonal) | Abcam | ab7817 | 1:100 dilution for IF |
| Antibody | GFP antibody (Chicken polyclonal) | Abcam | ab13970 | 1:100 dilution for IF |
| Commercial assay or kit | trueview autofluorescence quenching kit | Vector laboratories | SP-8400–15 | |
| Other | Alexa 594-conjugated WGA | ThermoFisher | W11262 | 1:200 dilution for IF |
| Software, algorithm | Pathology image analysis software Visiopharm | Visiopharm | | |
| Other | FITC-Dextran | Sigma-Aldrich | FD500S | |
| Other | alexa Fluor 555 Cadaverine | ThermoFisher | A30677 | |
| Other | Qdot 655 | ThermoFisher | Q21021MP | |
| Other | Hamilton neuros micro-injection syringe | Hamilton | 6546006 | |

*Continued on next page*

*Appendix 1—key resources table continued*

| Reagent type (species) or resource | Designation | Source or reference | Identifiers | Additional information |
|---|---|---|---|---|
| Antibody | PE-Cy7 Mouse Anti-Mouse CD45.2 antibody (Mouse monoclonal) | BD Pharmingen | 560696 | 1:100 dilution for FACS |
| Antibody | APC Rat Anti-Mouse CD31 Clone MEC 13.3 (RUO) antibody (Rat monoclonal) | BD Pharmingen | 561814 | 1:100 dilution for FACS |
| Other | DAPI | Thermo Fisher | D1306 | 1:5000 dilution for FACS |
| Commercial assay or kit | RNeasy MinElute cleanup kit | Qiagen | 74204 | |
| Commercial assay or kit | Smart-Seq ultra low input RNA kit | Takara | 634890 | |
| Commercial assay or kit | Nextera DNA XT Library Prep Kit | Illumina | FC-131–1096 | |
| Commercial assay or kit | NextSeq 500/550 High Output Kit | Illumina | 20024907 | |
| Software, algorithm | R | R-project | v.3.5.3 and v.3.6.1 | |
| Commercial assay or kit | U-plex chemokine combo 1 (Ms) kit | MSD | K15321K-1 | |
| Other | Siemens clinical analyzer | ADVIA Chemistry XPT system | | |
| Commercial assay or kit | BCP albumin assay kit | Sigma-Aldrich | MAK125-1KT | |
| Other | Taqman probes for human | Thermo Fisher | *LYVE1*, Hs00272659_m1; *PDPN*, Hs00366766_m1; *FLT4*, Hs01047677_m1; *VEGFC*, Hs01099203_m1; *HPRT1*, Hs02800695_m1 | |
| Software, algorithm | Graphpad Prism | Graphpad | | |

