## [Decision Letter]

**Acceptance summary:**

The manuscript shows the importance of lymphatic vasculature in chronic cardiac dysfunction with mechanical insights. Moreover, the authors elegantly show that lymphangiogenic therapy using VEGFCc156s can alleviate pathological changes and ameliorate heart function by improving cardiac lymphatic function and mitigating inflammation and hypertension. Finally, this study provide a potential therapeutic avenue for chronic cardiac dysfunction induced by hypertension.

**Decision letter after peer review:**

Thank you for submitting your article "Lymphangiogenic therapy prevents cardiac dysfunction by ameliorating inflammation and hypertension" for consideration by *eLife*. Your article has been reviewed by two peer reviewers, including Gou Young Koh as the Reviewing Editor and Reviewer #1, and the evaluation has been overseen by Matthias Barton as the Senior Editor.

The reviewers have discussed the reviews with one another and the Reviewing Editor has drafted this decision to help you prepare a revised submission. In the light of the comments made by the reviewers, we decided to offer a "major revision" prior to publishing your manuscript in *eLife*.

The comments of all two reviewers are in good agreement. While the reviewers found this work to be of importance as a translational research, they raised concerns about the lack of mechanistic insights on the preclinical VEGFC1563 therapy for chronic cardiac dysfunction. The authors are required to carefully address the comments point-by-point in a data-driven manner or with further analyses or discussions. Specifically, the authors are encouraged to pay attention to the major comment 1 of reviewer 1 and comments 1-3 of reviewer 2. If necessary, please provide the reasons for not implementing the suggested changes. We believe the authors could revise the manuscript successfully given their expertise.

Reviewer #1:

The manuscript by Song et al., shows the importance and mechanistic insights of lymphatic vasculature in chronic cardiac dysfunction. The authors demonstrated pathological and functional association of cardiac lymphatics in an angiotensin infusion-induced cardiac dysfunction model. Moreover, they showed that lymphangiogenic therapy using VEGFCc156s can alleviate pathological changes and ameliorate heart function by improving cardiac lymphatic function and mitigating inflammation and hypertension. The intriguing findings of this study suggest a potential therapeutic strategy for chronic cardiac dysfunction induced by hypertension. Therefore, I consider this work suitable for publication in *eLife* after addressing following points.

(1) VEGFC – VEGFR3 signaling pathway is the most well-known signaling axis regulating development and growth of lymphatic vasculature. The authors showed that VEGFCc156s infusion reduced the loss of cardiac lymphatic vasculature as demonstrated by increased podoplanin+ lymphatic vessel density and area in Figure 2J and Figure 2—figure supplement 1M. However, an interesting observation was made in kidney lymphatic vasculature, where the VEGFCc156s administration reduced lymphatic endothelial cell marker genes and lymphatic vessel density in Figure 6E, F and Figure 6—figure supplement 1B, C, D. What is the authors' explanation for such an opposite effect of VEGFCc156s, and what is the effect of VEGFCc156s on other organ lymphatics? For example, "skin" which is related to pathogenesis of hypertension.

2) How are cardiac lymphatic valves affected in AngII and AngII+VEGFC groups?

3) In Figure 2I and Figure 4G, what is the image showing? Is it ctrl, angll or angll+VEGFC group? The authors should provide a representative image for each group.

4) In Figure 4—figure supplement 1B, eGFP signal shown is quite not clear. Co-staining with LEC marker genes are required.

5) In Figure 2—figure supplement 1G and H – representative images for each group is required. In Figure 2—figure supplement 1L and Figure 6—figure supplement 1D, P-value for the quantification is omitted.

6) Results – LYVE1 is highly expressed in a subset of macrophages. How could the authors exclude the involvement of LYVE1 sequence from LYVE1+ macrophages when analyzing the SNP of the genome from whole human heart tissue?

7) Results – What is the source of VEGF-C?

8) Results – PDPN expression can be upregulated in inflamed heart, but the authors stated that the PDPN expression was reduced in chronic heart failure patients. What could be the reason for this discrepancy?

9) More defined analyses are required to visualize interstitial mononuclear cell infiltrate and tubular basophilia in the kidney.

10) Figure 1A can be arranged by Table 1.

11) Figure 2 – What is an implication of the lung weight change? Does it imply a reduction in lung edema?

12) In Figure 3 – The injection site and cardiac lymphatics are unclear.

13) Overall, the data presentation should be improved. It is hard to follow them.

Reviewer #2:

Song et al. report that continuous infusion of Vegfr-3 ligand VEGF-C156S prevents development of chronic cardiac dysfunction using angiotensin II (AngII) infusion-induced model. Continuous delivery of VEGFC156s increased heart lymphatic vascular density and lymphatic drainage, reduced AngII-induced cardiac and lung hypertrophy and improved cardiac contractibility together with a decreased Nppb expression The authors also analysed the transcriptional changes in LECs, BECs and cardiomyocytes isolated from the heart and diaphragm LECs. They report that AngII produces global changes at the transcriptomes of LECs, BECs and cardiomyocytes and many of the changes are resolved upon VEGFC156s, specifically in LECs of heart and diaphragm. AngII induced genes involved in inflammation and immune cell chemotaxis, changes that were reversed upon VEGFC156s treatment. They also suggest that TLR, Hif1a and Gli1 pathways drive LEC transcriptional changes in cardiac dysfunction, which are then ameliorated upon VEGF-C treatment. Since hypertension is one of the characteristics of chronic cardiac disease, the authors analysed the effect of VEGFC156s and demonstrate by that it reduces blood pressure. Finally, the authors report kidney dysfunction (by Urine volume and ACR ratio) recovery after VEGFC156s treatment.

Finding that lymphangiogenic therapy improves resolution of myocardial edema and inflammation, and prevents cardiac dysfunction and fibrosis is not new, as reported in several previous studies (e.g. Henri et al. and Houssari et al. ). However, here the authors convincingly show that co-administration of VEGF-C improves multiple pathological parameters in chronic cardiac dysfunction model. It is therefore a clinically relevant result, especially because the authors deliver the protein systemically and show that relatively modest two-fold increase in circulating levels of VEGF-C156S produces substantial therapeutic benefits. Unfortunately, the study is rather descriptive and the authors did not pursue further molecular analysis of cardiac and diaphragm lymphatic endothelial cells to better characterize the molecular changes imposed by failing heart and treatment with VEGF-C.

1) An important distinguishing feature of the current manuscript is systemic delivery of VEGFR-3 ligand, as opposed to cardiac delivery in previous publications. Therefore the authors should provide better characterization of the effects of systemic VEGF-C administration on lymphatic vasculature both in heart and other organs, such as skin. Notably, they should establish whether their infusion method leads to the expansion of initial capillary network or, perhaps enlargement of pre-collecting or collecting vessels. At least one publication demonstrated that recombinant VEGF-C156S promoted lymphatic contractility (Breslin et al., 2007), hence improved lymphatic draining function upon VEGF-C156S administration can be potentially uncoupled from its lymphangiogenic activity. Indeed, in the lymphoangiography images of Figure 3 the angII effects on the heart and the vessel leakiness are clear, however the Qdot images do not show much expansion of lymphatic vascular network. Furthermore, increased skin lymphangiogenesis has been shown to improve outcomes in salt-induced hypertension model (Machnik et al., 2009), therefore lymphatic vessel expansion in organs other than heart is potentially directly relevant.

2) The beneficial effects of VEGFC156S infusion in AngII-infusion model are encouraging. However, AngII and VEGF-C156S were infused simultaneously. To know if lymphangiogenic treatment can be used as therapy, ideally it would be important to know whether VEGFC-156S administration helps after establishment of cardiac dysfunction. In the very least, the timeline of cardiac and other organ lymphangiogenesis should be documented in more detail.

3) Do the authors check the effect of VEGFC156s alone? It would have been great added value to include it as control in all in vivo experimentation.

4) The authors made a number of broad conclusions based on bulk RNAseq data of cardiac and diaphragm LECs. However, I am not convinced that they monitor cell-autonomous effects in their analyses. For example, several genes that they report to be modulate in LECs in their models are to the best of my knowledge, expressed in immune cells and not in LECs, such as PF4, ITGAM, CD276 etc. One potential explanation, and we observed this effect also in our models, is that bulk sorted LECs can be contaminated with small number of immune cells (and of course other cell types, depending on the organ). Please provide evidence of the purity of isolated populations by analysing the expression of established markers, such as CD31 and CDH5 (endothelial identity), *Prox1*, *Flt1* and *Nrp1* (lymphatic and blood endothelial cell markers), *Ptprc* (immune cells), collagen 6 subunits (fibroblasts) and confirm that at least some of the functionally relevant transcripts, identified as differentially expressed in LECs by bioinformatics analyses, are indeed expressed in lymphatic endothelial cells by immunostaining. Similarly, the suggestions that N-Myc drives VEGF-C effects and that AngII induces and Vegf-C repress activation of TLR, Hif1A and Gli1 pathways in LECs, are intriguing, but they are not substantiated by any other means, and therefore remains speculative.

5) Since patients with hypertension harbour lower VEGFC serum levels, did the authors analysed what were the serum levels of endogenous VEGF-C in their hypertension model for angII and angII+ VEGFC156s conditions?

6) The authors show that in concordance with reduction of hypertension, VEGFC156s improved kidney dysfunction. But what is the effect of VEGFC156s in renal lymphatics? Only expression levels of lymphatic markers and density analysis was performed. The authors confirm previous studies showing that in response to hypertension there is expansion of lymphatic vessels in kidneys and observe a decrease LEC markers expression upon VEGFC156s. This part is controversial, how do the authors explain the decrease on lymphatic markers expression upon VEGF156s? The authors suggest this is a result of the low dose used, however this dose was enough to recover lymphatic function, inflammation and hypertension and poor retention in the kidney would justify no differences, however there is a decrease of lymphatic endothelial cell markers when comparing angII and angII+ VEGFC. Indeed, in the kidney the effect of angII and angII +VEGFC156s on lymphatic markers is opposite to what is observed in the heart. How do the authors explain this point?

7) AngII infusion is known to induce extensive interstitial fibrosis. What was the effect of VEGF-C156S administration on this important pathophysiological parameter?

---

## [Author Response]

Reviewer #1:[…] (1) VEGFC – VEGFR3 signaling pathway is the most well-known signaling axis regulating development and growth of lymphatic vasculature. The authors showed that VEGFCc156s infusion reduced the loss of cardiac lymphatic vasculature as demonstrated by increased podoplanin+ lymphatic vessel density and area in Figure 2J and Figure 2—figure supplement 1M. However, an interesting observation was made in kidney lymphatic vasculature, where the VEGFCc156s administration reduced lymphatic endothelial cell marker genes and lymphatic vessel density in Figure 6E, F and Figure 6—figure supplement 1B, C, D. What is the authors' explanation for such an opposite effect of VEGFCc156s, and what is the effect of VEGFCc156s on other organ lymphatics? For example, "skin" which is related to pathogenesis of hypertension.

We agree with the reviewer that systemic administration of VEGFCc156s could have a multitude of effects outside of the heart. To address this concern, we examined the effect of VEGFCc156s on ear skin lymphatic vessels 6 weeks post-angiotensin II infusion, and observed that similar to the effects of VEGFCc156s on the lymphatic vessels in heart at this timepoint, VEGFCc156s treatment significantly increased ear skin lymphatic vessel diameter and lymphatic vascular density, supporting a systemic lymphangiogenic response induced by VEGFCc156s treatment (new data in Figure 2—figure supplement 1O-R). We agree that the changes in skin lymphatics could contribute to the effect of the treatment on hypertension and we have added text to the Discussion accordingly.

We agree with the reviewer that the lymphatic marker qPCR and *lyve1* staining data in the kidney at the 6-week timepoint are intriguing. As new data with this revision, we conducted in vivo studies to profile the time-course of the protective effects of VEGFCc156s on angiotensin II-induced cardiac dysfunction and characterized the changes induced by the treatment at 1 week after treatment. In this new study, VEGFCc156s treatment in healthy and angiotensin II-infused animals induced a trend towards increased lymphatic vessel density in skin and a significant increase in podoplanin-positive lymphatic vessel density in heart, indicating that VEGFCc156s can indeed induce lymphangiogenesis (Figure 7—figure supplement 1I-L). However, at this time point, *Lyve1* expression and *Lyve1*-positive lymphatic vessel density was not altered in the kidney of animals treated with VEGFCc156s alone (Figure 7—figure supplement 1F-H), confirming that the VEGFCc156s treatment did not induce lymphangiogenesis in kidney at the dose we use for our studies. These new data along with the data on kidney lymphatics at 6 weeks after angiotensin II infusion (Figure 6E, F, Figure 6—figure supplement 1E-H) suggest that the increase in renal *lyve1* gene expression and lymphatic vessel density in this model could be secondary to the hypertension status of the animals. At 1 week, the hypertension is not sufficient to drive this change, and by 6 weeks, hypertension has caused an adaptation in angiotensin II-infused animals to induce lymphangiogenesis that is ameliorated by the reduced hypertension in VEGFCc156s-infused, angiotensin II-treated animals. In addition, it is possible that the systemic infusion of VEGFCc156s did not significantly affect kidney lymphatics due to the low dose and poor local kidney retention, whereas transgenic overexpression of VEGFD specifically in kidney as in previous reports would sustain high local expression and therefore produce a greater local effect. We have discussed these hypotheses in the Discussion section.

2) How are cardiac lymphatic valves affected in AngII and AngII+VEGFC groups?

Unfortunately, we made cross-sections of the heart for immunofluorescence staining and we did not have additional heart tissue to perform whole mount imaging of the lymphatic valves. Therefore, we were unable to evaluate this parameter. This is discussed as a limitation in the manuscript Discussion and will be an area of interest for us to investigate in the future.

3) In Figure 2I and Figure 4G, what is the image showing? Is it ctrl, angll or angll+VEGFC group? The authors should provide a representative image for each group.

We have now added representative images for podoplanin and smooth muscle actin staining for each treatment group in Figure 2I as well as representative images of CD68, Smooth muscle actin and DAPI staining for each treatment group in Figure 4G.

4) In Figure 4—figure supplement 1B, eGFP signal shown is quite not clear. Co-staining with LEC marker genes are required.

We are pleased to provide new immunofluorescence staining data using mouse heart cross sections and whole mount staining using diaphragm tissue from *Prox1*-eGFP mice with GFP and *Lyve1* co-staining (Figure 4—figure supplement 1C). The GFP stain overlapped with the *Lyve1* stain, indicating that the eGFP signal came from the lymphatic vessels in the *Prox1*-eGFP mice.

5) In Figure 2—figure supplement 1G and H – representative images for each group is required. In Figure 2—figure supplement 1L and Figure 6—figure supplement 1D, P-value for the quantification is omitted.

We have added representative WGA stain images from heart cross-sections in Figure 2—figure supplement 1H. We have also added the p-value for the quantifications previously shown in Figure 2—figure supplement 1L and Figure 6—figure supplement 1D (in the current version, the figures are Figure 2—figure supplement 1N and Figure 6—figure supplement 1H).

6) Results – LYVE1 is highly expressed in a subset of macrophages. How could the authors exclude the involvement of LYVE1 sequence from LYVE1+ macrophages when analyzing the SNP of the genome from whole human heart tissue?

We appreciate this helpful comment from the reviewer. The associations we observed in GWAS analysis are the associations between a specific loci with a disease phenotype where *Lyve1* and another gene are located or close to, and the results do not indicate the exact gene that mediates the effect or the exact cell type that mediates the effect. We agree with the reviewer that it is possible that several types of cells including both *Lyve1*+ macrophages and *Lyve1*+ lymphatic vessels may mediate the effect, and unfortunately the GWAS analysis will not provide us with the information concerning cell types due to the nature of the analysis. We have revised our description on the GWAS data to make it more accurate (Discussion).

7) Results – What is the source of VEGF-C?

The VEGFCc156s we used for our in vivo studies is the recombinant human VEGF-C (Cys156Ser) protein purchased from R&D systems. It has been used previously to induce lymphangiogenesis in vivo in mice (Klotz et al., 2015), and the sequence of this human VEGFC short-form is identical to the mouse VEGFC short-form.

8) Results – PDPN expression can be upregulated in inflamed heart, but the authors stated that the PDPN expression was reduced in chronic heart failure patients. What could be the reason for this discrepancy?

We are aware of previous publications (Cimini, et al., 2017; Miguel Quintanilla, et al., MDPI, 2019; Cimini, et al., 2019) indicating that podoplanin expression could be upregulated in inflamed hearts due to an increase in podoplanin-expressing cells that are not lymphatic endothelial cells. This upregulation would be most significant in the highly inflamed/fibrotic infarcted area due to the local accumulation of podoplanin-expressing cells that facilitate local inflammation. To that end, we used chronic heart failure human hearts and a chronic cardiac dysfunction model, which have a very different pathogenesis compared with a myocardial infarction model, which represents acute injury. In addition, compared with the previous reports that looked specifically at the infarcted/fibrotic site in heart, we sampled from the whole heart and also utilized immunohistochemistry staining for podoplanin-positive lymphatic vessels, in which we excluded podoplanin-positive cells that were not part of a vessel from the vessel quantification using a size filter in our visiopharm-based image analysis (subsection “Histology and immunohistochemistry”). Thus, we suggest that the changes in podoplanin expression we observed in our disease model largely come from the lymphatic vasculature but not podoplanin-expressing cells, and the discrepancy between our findings and previous reports is due to the differences in disease models and disease pathogenesis. We have added the references into our revised manuscript and added a Discussion point concerning this.

9) More defined analyses are required to visualize interstitial mononuclear cell infiltrate and tubular basophilia in the kidney.

We appreciate the comment from the reviewer. The pathologist that evaluated the H&E staining in the kidney used this nomenclature to describe the pathological findings. To improve visualization of these endpoints, we first enlarged the H&E image in Figure 6—figure supplement 1A to improve the image resolution. To further evaluate kidney pathology, we conducted a PSR stain using the same kidney sections to add an additional evaluation of kidney fibrosis. The results indicated that angiotensin II infusion induced renal fibrosis, which was significantly reduced by VEGFCc156s treatment (Figure 6—figure supplement 1A, C, D). We feel that the new PSR staining results will strengthen the kidney datasets reported at 6-week post-angiotensin II infusion and our overall conclusions.

10) Figure 1A can be arranged by Table 1.

We were informed by the editor that the format of table supplements is not preferred by the journal and therefore we have rearranged Figure 1 and converted the original Figure 1A (GWAS analysis) into Figure 1—figure supplement 1.

11) Figure 2 – What is an implication of the lung weight change? Does it imply a reduction in lung edema?

An increase in lung weight is commonly observed in severe cardiac dysfunction or heart failure and it does reflect a change in lung edema (King and Goldstein, 2020). We regularly use both heart weight and lung weight to evaluate cardiac function of the animals at euthanasia, and it’s possible that the reduction in the lung weight of VEGFCc156s-treated, angiotensin II-infused animals compared with animals treated with angiotensin II alone implied a reduction in lung edema. However, definitive measurement of lung edema requires measurement of the wet/dry ratio using fresh lungs and unfortunately, we no longer have fresh lungs to perform this measurement. We have added text to denote this possibility to the Results section.

12) In Figure 3 – The injection site and cardiac lymphatics are unclear.

We have revised Figure 3—figure supplement 1 and added a red arrowhead to the injection site (apex) and have used white arrow heads to delineate the cardiac lymphatics.

13) Overall, the data presentation should be improved. It is hard to follow them.

We appreciate that the complex data package has made the data somewhat difficult to follow. With this revision, we have added new data and refined the data presentation. We have additionally carefully edited the manuscript for additional clarity. We feel that these revisions have improved the data flow, and we thank the reviewer for calling this to our attention.

Reviewer #2:[…] Finding that lymphangiogenic therapy improves resolution of myocardial edema and inflammation, and prevents cardiac dysfunction and fibrosis is not new, as reported in several previous studies (e.g. Henri et al. and Houssari et al. ). However, here the authors convincingly show that co-administration of VEGF-C improves multiple pathological parameters in chronic cardiac dysfunction model. It is therefore a clinically relevant result, especially because the authors deliver the protein systemically and show that relatively modest two-fold increase in circulating levels of VEGF-C156S produces substantial therapeutic benefits. Unfortunately, the study is rather descriptive and the authors did not pursue further molecular analysis of cardiac and diaphragm lymphatic endothelial cells to better characterize the molecular changes imposed by failing heart and treatment with VEGF-C.1) An important distinguishing feature of the current manuscript is systemic delivery of VEGFR-3 ligand, as opposed to cardiac delivery in previous publications. Therefore the authors should provide better characterization of the effects of systemic VEGF-C administration on lymphatic vasculature both in heart and other organs, such as skin. Notably, they should establish whether their infusion method leads to the expansion of initial capillary network or, perhaps enlargement of pre-collecting or collecting vessels. At least one publication demonstrated that recombinant VEGF-C156S promoted lymphatic contractility (Breslin et al., 2007), hence improved lymphatic draining function upon VEGF-C156S administration can be potentially uncoupled from its lymphangiogenic activity. Indeed, in the lymphoangiography images of Figure 3 the angII effects on the heart and the vessel leakiness are clear, however the Qdot images do not show much expansion of lymphatic vascular network. Furthermore, increased skin lymphangiogenesis has been shown to improve outcomes in salt-induced hypertension model (Machnik et al., 2009), therefore lymphatic vessel expansion in organs other than heart is potentially directly relevant.

We appreciate this constructive comment from the reviewer and have significant additional data to include to address this point in the manuscript revision. To address this comment, we first conducted an additional analysis on the podoplanin-positive lymphatic vessel quantification in heart to evaluate the average podoplanin-positive lymphatic vessel lumen area. Indeed, as the reviewer pointed out, there was a significant enlargement of the podoplanin-positive pre-collector vessel lumen in VEGFCc156s-treated angiotensin II-infused hearts compared with hearts treated with angiotensin II alone (Figure 2K), which was consistent with the significant improvement we observed in cardiac lymphatic transport in VEGFCc156s-treated hearts using lymphangiography (Figure 3) and indicated a functional change in the collecting vessels.

In addition, we examined the effect of VEGFCc156s on the lymphatic vessels in ear skin 6 weeks after treatment. Similar to the effects of VEGFCc156s on the lymphatic vessels in heart at this timepoint, VEGFCc156s treatment significantly increased lymphatic vessel diameter and lymphatic vascular density in the ear skin of angiotensin II-infused animals, supporting a systemic lymphangiogenic response induced by VEGFCc156s treatment (Figure 2—figure supplement 1O-R). We agree that the changes in the lymphatics in skin induced by VEGFCc156s treatment could contribute to the effect of the treatment on hypertension, and we have added a point on this in the Discussion.

2) The beneficial effects of VEGFC156S infusion in AngII-infusion model are encouraging. However, AngII and VEGF-C156S were infused simultaneously. To know if lymphangiogenic treatment can be used as therapy, ideally it would be important to know whether VEGFC-156S administration helps after establishment of cardiac dysfunction. In the very least, the timeline of cardiac and other organ lymphangiogenesis should be documented in more detail.

We agree with the reviewer that a therapeutic reversal of the cardiac disease using VEGFCc156s would be a very compelling experiment. However, due to the minipump administration of angiotensin II and VEGFCc156s, this is very difficult to achieve, as our IACUC limits the number of surgical procedures on mice (only one surgery per animal allowed). Therefore unfortunately, we are unable to conduct a reversal experiment using the infusion method. We also agree with the reviewer that a more careful assessment of the lymphangiogenesis timeline should be performed. Following this suggestion, we conducted in vivo studies to characterize the timecourse of the beneficial effect of VEGFCc156s on angiotensin II-induced cardiac dysfunction. We specifically examined the effect of VEGFCc156s 1-week post-infusion, at which time, the blood pressure was equally high in both angiotensin II-infused and VEGFCc156s-treated angiotensin II-infused mice based on the telemetry study result from Figure 5. We found that at this early time point, whereas VEGFCc156s does not have an effect on blood pressure, it significantly reduced angiotensin II-induced gene expression in pathways involved in cardiac hypertrophy, fibrosis, and inflammation. Furthermore, VEGFCc156s infusion alleviated angiotensin II-induced cardiac fibrosis and inflammation as assessed by histological staining at this early 1-week timepoint. These new results are included in Figure 7 and Figure 7—figure supplement 1 and 2. We feel that these new data significantly improve the quality of the manuscript and solidify the conclusions drawn. We thank the reviewer for the constructive advice.

3) Do the authors check the effect of VEGFC156s alone? It would have been great added value to include it as control in all in vivo experimentation.

We agree that a VEGFCc156s alone arm would have added value in our studies. Following this advice, we added a treatment arm in new data added here that assesses one week of infusion of VEGFCc156s alone. In this assessment, a lymphangiogenic response in heart was observed in VEGFCc156s-alone infused animals, which was similar to animals that were infused with VEGFCc156s and angiotensin II concurrently. These new results are included in Figure 7 and Figure 7—figure supplement 1 and 2.

4) The authors made a number of broad conclusions based on bulk RNAseq data of cardiac and diaphragm LECs. However, I am not convinced that they monitor cell-autonomous effects in their analyses. For example, several genes that they report to be modulate in LECs in their models are to the best of my knowledge, expressed in immune cells and not in LECs, such as PF4, ITGAM, CD276 etc. One potential explanation, and we observed this effect also in our models, is that bulk sorted LECs can be contaminated with small number of immune cells (and of course other cell types, depending on the organ). Please provide evidence of the purity of isolated populations by analysing the expression of established markers, such as CD31 and CDH5 (endothelial identity), Prox1, Flt1 and Nrp1 (lymphatic and blood endothelial cell markers), Ptprc (immune cells), collagen 6 subunits (fibroblasts) and confirm that at least some of the functionally relevant transcripts, identified as differentially expressed in LECs by bioinformatics analyses, are indeed expressed in lymphatic endothelial cells by immunostaining. Similarly, the suggestions that N-Myc drives VEGF-C effects and that AngII induces and Vegf-C repress activation of TLR, Hif1A and Gli1 pathways in LECs, are intriguing, but they are not substantiated by any other means, and therefore remains speculative.

We appreciate the reviewer’s concern regarding the purity of the cell isolations used for RNAseq. We used several methods to ensure purity of sorted cells. First, we digested the hearts and did a slow spin to isolate heavy cells such as cardiomyocytes. We then flow sorted for CD45.2 (encoded by *Ptprc*) negative cells, which should ensure that we are not identifying immune cells, and we set stringent gates to ensure the sorted cell populations were expressing the correct markers (cell sorting workflow and gating parameters are detailed in Figure 4—figure supplement 1D). We conducted multiple tests to optimize the cell isolation workflow and performed qPCR to assess various marker genes in the sorted cell populations to assess purity. We provide a figure to the reviewer here from one of our test runs (Author response image 1) demonstrating that the cardiomyocyte population (isolated by low speed centrifugation, which is relatively impure and may contain other cell types in addition to cardiomyocytes) expressed high levels of cardiac markers (*Myh7*, *Actn2*), whereas endothelial cells (sorted by CD45.2 negative, CD31 positive) did not express *Ptprc* (CD45) and expressed *Pecam1* (CD31). In addition, lymphatic endothelial cell populations (sorted by CD45.2 negative, CD31 positive, GFP positive) expressed high levels of lymphatic endothelial markers (*Lyve1*, *Pdpn*, *Flt4* and *Prox1*) (shown in Author response image 1).

**Author response image 1. respfig1:** Analysis of cell-specific marker expression in isolated cardiomyocytes (CMs), cardiac lymphatic endothelial cells (ECs), cardiac blood ECs and diaphragm lymphatic ECs. The cells were isolated from the heart and diaphragm of a *Prox1*-eGFP mouse following the cell isolation procedure shown in Figure 4—figure supplement 1D in main manuscript. RNA was isolated from the cells, and quantitative RT-PCR was performed for *Myh7*, *Actn2*, *Ptprc*, *Pecam1*, *Lyve1*, *Prox1*, *Flt4*, *Pdpn* and normalized to the expression of housekeeping gene *Hprt*(*n*=1/group).

For our RNAseq analysis, a PCA analysis indicated that each isolated cell population clustered together and showed distinct expression profiling (Figure 4—figure supplement 1E). In addition, consistent with our pilot optimization test result, our RNAseq analysis could not detect *ptprc* (CD45) expression and detected high counts of *Pecam1* (CD31) (TPM count 410-530 for lymphatic ECs, 740-850 for blood ECs and 137-138 for cardiomyocytes) and Cdh5 (TPM count 65-90 for lymphatic ECs, 164-220 for blood ECs and 22-29 for cardiomyocytes) in our isolated blood and lymphatic endothelial cell populations. Moreover, the lymphatic endothelial cell populations showed high counts/expression of *lyve1*(TPM count 1300-4200), *Prox1* (TPM count 159-254), *Pdpn* (TPM count 63-131) and *Flt4* (TPM count 428-1073) compared with other cell populations (*Lyve1* TPM count 17-51, *Prox1* TPM count 1-5, *Pdpn* TPM count 2-6, *Flt4* TPM count 8-57 for BECs and cardiomyocytes). The cardiomyocyte population (isolated by centrifugation, relatively impure) showed a relatively high expression of collagen 6 subunits (Col6a1 TPM count 100-210, Col6a2 TPM count 100-220) while lymphatic endothelial cell populations showed a low expression of collagen 6 subunits (Col6a1 TPM count 10-34, Col6a2 TPM count 12-27). We could only isolate 800-1200 lymphatic endothelial cells/heart and 1000-2000 lymphatic endothelial cells/diaphragm with the stringent cell isolation protocol, and the cell amount was just enough for a low-input RNAseq method for the transcriptional profiling of the distinct cell populations. Therefore unfortunately, we don’t have extra cells for additional immunostaining tests to confirm the identity of each isolated cell population. However, based on our stringent isolation and gating procedures described above as well as the RNAseq TPM count data, we suggest that the cell populations we isolated for RNAseq were relatively pure. We agree with the reviewer that the RNAseq results and pathway analysis need to be validated with more experiments, and that will fall into the scope of future studies.

5) Since patients with hypertension harbour lower VEGFC serum levels, did the authors analysed what were the serum levels of endogenous VEGF-C in their hypertension model for angII and angII+ VEGFC156s conditions?

VEGFC concentration was assessed in plasma of the animals from our in vivo studies at both the 1-week and 6-week timepoints, and the results are included in Figure 2—figure supplement 1B and Figure 7—figure supplement 1A. There was no significant difference in serum VEGFC levels between ctrl animals ang angiotensin II-infused animals. However, the human study we cited in the Discussion indicated that low serum VEGFC level was associated with a higher risk of hypertension, which is consistent with our finding that increasing serum VEGFC level could be beneficial for hypertension. In addition, the data from the human study we cited were from patients with refractory hypertension that doesn’t respond to regular anti-hypertensive therapy and there could be differences in pathogenesis and disease mechanisms between refractory hypertension in human and our angiotensin II model. We agree that further investigations are needed to understand the detailed mechanisms underlying these observations.

6) The authors show that in concordance with reduction of hypertension, VEGFC156s improved kidney dysfunction. But what is the effect of VEGFC156s in renal lymphatics? Only expression levels of lymphatic markers and density analysis was performed. The authors confirm previous studies showing that in response to hypertension there is expansion of lymphatic vessels in kidneys and observe a decrease LEC markers expression upon VEGFC156s. This part is controversial, how do the authors explain the decrease on lymphatic markers expression upon VEGF156s? The authors suggest this is a result of the low dose used, however this dose was enough to recover lymphatic function, inflammation and hypertension and poor retention in the kidney would justify no differences, however there is a decrease of lymphatic endothelial cell markers when comparing angII and angII+ VEGFC. Indeed, in the kidney the effect of angII and angII +VEGFC156s on lymphatic markers is opposite to what is observed in the heart. How do the authors explain this point?

We agree with the reviewer that the lymphatic marker qPCR and *lyve1* staining data in the kidney at the 6-week timepoint are intriguing. As new data with this revision, we conducted in vivo studies to profile the time-course of the protective effects of VEGFCc156s on angiotensin II-induced cardiac dysfunction and characterized the changes induced by the treatment at 1 week after treatment. In this new study, VEGFCc156s treatment in healthy and angiotensin II-infused animals induced a trend towards increased lymphatic vessel density in skin and a significant increase in podoplanin-positive lymphatic vessel density in heart, indicating that VEGFCc156s can indeed induce lymphangiogenesis (Figure 7—figure supplement 1I-L). However, at this time point, *Lyve1* expression and *Lyve1*-positive lymphatic vessel density was not altered in the kidney of animals treated with VEGFCc156s alone (Figure 7—figure supplement 1F-H), confirming that the VEGFCc156s treatment did not induce lymphangiogenesis in kidney at the dose we use for our studies. These new data along with the data on kidney lymphatics at 6 weeks after angiotensin II infusion (Figure 6E, F, Figure 6—figure supplement 1E-H) suggest that the increase in renal *lyve1* gene expression and lymphatic vessel density in this model could be secondary to the hypertension status of the animals and not a direct effect of VEGFCc156s on the kidney. At 1 week, the hypertension is not sufficient to drive this change, and by 6 weeks, hypertension has caused an adaptation in angiotensin II-infused animals to induce lymphangiogenesis that is ameliorated by the reduced hypertension in VEGFCc156s-infused, angiotensin II-treated animals. In addition, it is possible that the systemic infusion of VEGFCc156s did not significantly affect kidney lymphatics due to the low dose and poor local kidney retention, whereas transgenic overexpression of VEGFD specifically in kidney as in previous reports would sustain high local expression and therefore produce a greater local effect. We have discussed these hypotheses in the Discussion section.

7) AngII infusion is known to induce extensive interstitial fibrosis. What was the effect of VEGF-C156S administration on this important pathophysiological parameter?

We are pleased to provide significant new data to address this point. PSR staining of mouse kidney sections indicated angiotensin II infusion induced significant kidney fibrosis after 6 weeks of infusion, which was significantly reduced in VEGFCc156s-treated animals(Figure 6—figure supplement 1A, C, D), indicating a beneficial effect of VEGFCc156s on kidney fibrosis and supporting an overall beneficial effect of VEGFCc156s on angiotensin II-induced kidney dysfunction at this timepoint. In addition, we also observed that VEGFCc156s treatment significantly reduced angiotensin II-induced pro-fibrotic gene expression and trended to reduce the level of cardiac fibrosis in the heart as early as 1-week post-treatment (Figure 7I-L, Figure 7—figure supplement 1P, Q). These new results indicate that VEGFCc156s improved the cardiac function in angiotensin II-infused animals by alleviating cardiac fibrosis in addition to the other mechanisms proposed in this manuscript. We thank the reviewer for this suggestion.